# A Greedy PDE Router for Blending Neural Operators and Classical Methods

## Abstract

When solving PDEs, classical numerical solvers are often computationally expensive, while machine learning methods can suffer from spectral bias, failing to capture high-frequency components. Designing an optimal hybrid iterative solver–where, at each iteration, a solver is selected from an ensemble of solvers to leverage their complementary strengths–poses a challenging combinatorial problem. While the greedy selection strategy is desirable for its constant-factor approximation guarantee to the optimal solution, it requires knowledge of the true error at each step, which is generally unavailable in practice. We address this by proposing an approximate greedy router that efficiently mimics a greedy approach to solver selection. Empirical results on the Poisson and Helmholtz equations demonstrate that our method outperforms single-solver baselines and existing hybrid solver approaches, such as HINTS, achieving faster and more stable convergence.

## 1 Introduction

Natural phenomena and engineered systems are often governed by ordinary and partial differential equations (PDEs). Solving these equations enables a variety of tasks - predicting the evolution of the system through simulation (e.g., forecasting weather using the Navier-Stokes equations) (Kalnay, 2003; Bauer et al., 2015; Staniforth & Côté, 1991), addressing control problems (e.g., optimizing heat shield design for spacecrafts using heat transfer equations) (Anderson, 1989; Tröltzsch, 2010), and tackling inverse problems based on external measurements (e.g., reconstructing brain activity from EEG data using electrophysiological models) (Baillet et al., 2001; Grech et al., 2008).

Traditionally, PDEs are solved using finite difference methods (Smith, 1985; LeVeque, 2007), which discretize the spatial and/or temporal domain and approximate the solution by solving a system of equations at the discrete grid points. Similarly, finite element methods (Hughes, 2003; Bathe, 2006; Brenner, 2008) construct a system of equations based on fitted curves for each finite element, and then rely on numerical linear algebra techniques–such as the Jacobi and Gauss-Seidel method (Saad, 2003)–to compute the solution. Such iterative methods, however, lack generalization across different initial conditions, boundary conditions, or forcing functions, as even minor changes to any of these parameters require re-solving the entire system of equations from scratch.

These challenges have motivated the development of neural operators (Kovachki et al., 2023)–a class of machine learning models that aim to learn the solution operator directly, enabling fast inference across a range of parameters. Neural operators lift the classical linear layer in neural networks to a linear operator–typically a kernel integral operator–between infinite-dimensional function spaces. While the universal approximation theorem of operators (Chen & Chen, 1995) suggests that neural operators can approximate solution operators with high accuracy, they are prone to spectral bias (Liu & Cai, 2021; Liu et al., 2024; You et al., 2024; Khodakarami et al., 2025; Xu et al., 2025)–similar to traditional neural networks (Rahaman et al., 2019; Xu et al., 2019; Luo et al., 2019)–and favor learning the low-frequency components of the solution function, often struggling to capture high-frequency features that iterative solvers excel at.

Recognizing this complementarity, Zhang et al. (2024) proposed HINTS, a hybrid solver that interleaves a neural operator pass every $\tau^{\text{th}}$ iteration (with $\tau$ fixed a priori) of a classical solver. While HINTS reports significant empirical gains in convergence and accuracy over the standalone neural operator and Jacobi solver, this fixed schedule can be detrimental: a poorly timed neural correction may increase the error and undo recent progress.

Although dynamic solver schedules are desirable, identifying the optimal sequence of solvers that minimizes final error is combinatorial since the search space grows exponentially with the number of iterations. A practical alternative is the greedy rule that, at each iteration, selects the solver with the largest immediate error reduction. Such a rule can be near-optimal when the final error satisfies (weak) supermodularity–so that applying a beneficial solver earlier is at least as valuable as applying it later. To support this approach, we make the following contributions:

- We present a general hybrid PDE solver in which a routing rule chooses, at each iteration, from a set of classical solvers and neural operators (not just two). With oracle access to the true error at every step, we show that a greedy routing rule achieves a constant-factor approximation to the optimal strategy for linear PDEs, provided the updates are error-reducing (Lipschitz with constant $< 1$) and zero-preserving–conditions under which weak supermodularity holds.

- Given that the true error is not observed at test time, a convex surrogate loss is introduced, which, when minimized, enables the learned model to imitate the greedy solver without access to true error information. This alignment is guaranteed through Bayes consistency.

- We empirically demonstrate that our approximate greedy solver outperforms HINTS and individual solvers in terms of faster, more stable convergence on benchmark PDEs, such as the Poisson and Helmholtz equations.

## 2 PROBLEM SETTING AND BACKGROUND

Consider the following linear PDE:

$$\mathcal{L}_x^a(u) = f, x \in D; \quad \mathcal{B}_x(u) = b, x \in \partial D \tag{1}$$

where $\mathcal{L}_x^a$ is a differential operator with respect to the spatial variable $x \in D$ parameterized by the coefficient function $a$, $f$ is a forcing function, $u$ is the solution to the PDE, $\mathcal{B}_x$ is the boundary operator, and $b$ is the boundary term. Such PDEs can be solved numerically using various discretization strategies, such as finite differences (Smith, 1985; LeVeque, 2007), finite element, and spectral methods (Boyd, 2001; Canuto et al., 2006). Here, we focus on finite differences, which replace derivatives with difference quotients (e.x., $\partial_x u(x) \approx (u(x + h) - u(x))/h$) on a uniform grid.

Formally, let $h$ be the grid size and $G_h(D)$ be a uniform grid with spacing $h$ over $D$. Given a function $g : D \to \mathbb{R}$, we represent its restriction to $G_h(D)$ by $g_h \in \mathbb{R}^N$, where $N = |G_h(D)|$. If $\mathcal{L}_h^a \in \mathbb{R}^{N \times N}$ is the discretized operator, the discrete counterpart of Equation (1) is expressed as

$$\mathcal{L}_h^a u_h = f_h, \tag{2}$$

with the boundary conditions incorporated in $\mathcal{L}_h^a$. Thus, the PDE reduces to a linear system of equations. Unlike much of the neural operator literature, which retains function space formulations for discretization invariance, we follow the conventions of classical numerical analysis and represent all functions and operators as finite-dimensional vectors and matrices. Discretization invariance is not essential here, since we focus on fixed-grid problems.

### 2.1 ITERATIVE PDE SOLVERS

Direct methods like Gauss Elimination and Thomas Algorithm can be computationally expensive in high-dimensional domains when solving systems like Equation (2). In contrast, iterative methods like Jacobi and Gauss-Seidel offer computational speedups by iteratively updating the solution, gradually converging to the true solution for the PDE. The general iterative update is

$$u^{(t+1)} = u^{(t)} + C\left(f_h - \mathcal{L}_h^a u^{(t)}\right), \tag{3}$$

where $u^{(t)}$ is the $t^{\text{th}}$ iterate of the solution and $C$ is a preconditioning matrix. Jacobi uses $C = D^{-1}$, where $D$ is the diagonal of $\mathcal{L}_h^a$, and Gauss-Seidel uses $C = (D + L)^{-1}$, where $L$ denotes the strict lower triangular part. However, iterative methods tend to damp low frequency components of the error slowly. Multigrid solvers (Briggs et al., 2000; Trottenberg et al., 2001; Hackbusch, 2013) tackle this problem by alternating smoothing on coarse and fine grid resolutions to dampen more uniformly across frequencies. We defer the reader to Appendix A.1 for more background on Multigrid solvers.

## 2.2 Neural Operators

Suppose we observe samples $\{(a_i, f_i, u_i)\}_{i=1}^N$ such that $(a_i, f_i) \sim \mathcal{P}$ are i.i.d. samples drawn from some distribution. Let $u_i$ be generated by a deterministic solution operator $\mathcal{G}^*$, i.e., $u_i = \mathcal{G}^*(a_i, f_i)$. A neural operator $\mathcal{G}_\theta$ seeks to approximate $\mathcal{G}^*$ by minimizing the expected squared $L^2(D)$ error (with respect to the Lebesgue measure): $\mathcal{R}_{\mathrm{NO}}(\mathcal{G}_\theta) = \mathbb{E}_{(a,f) \sim \mathcal{P}} [l_{\mathrm{NO}}(a, f, u, \mathcal{G}_\theta)]$, where

$$l_{\mathrm{NO}}(a_i, f_i, u_i, \mathcal{G}_\theta) = \int_D \|u_i(x) - \mathcal{G}_\theta(a_i, f_i)(x)\|_2^2 \, dx.$$

Neural operators often use discretization-invariant layers. Prominent instantiations include Fourier Neural Operators, which use spectral transforms (Li et al., 2020a), Graph Neural Operators, which perform graph-based aggregation over sampled points (Li et al., 2020b), and DeepONets, which employ a trunk–branch decomposition to map input functions to output functions (Lu et al., 2021).

## 2.3 Greedy Optimization

Consider the problem of maximizing $g(S)$ over $S \subseteq \Omega$ with $|S| \leq T$ where $g : 2^\Omega \to \mathbb{R}$ is a set function defined on subsets of a set $\Omega$. An exhaustive search over all subsets quickly becomes infeasible. An efficient alternative is the greedy algorithm. It builds the solution subset iteratively by, at each step, adding the element $\omega^*$ that yields the largest marginal gain, i.e. $\omega^* = \arg\max_{\omega \in \Omega \setminus S} g(S \cup \omega)$, and updating $S \leftarrow S \cup \{\omega^*\}$. When $g$ is non-negative, monotone (adding elements never decreases the value), and submodular (diminishing returns), the greedy algorithm achieves a $(1 - e^{-1})$ approximation to the optimal solution (Nemhauser et al., 1978). Formally, a function $g$ is submodular if $g(A \cup \{\omega\}) - g(A) \geq g(B \cup \{\omega\}) - g(B)$ for all $A \subseteq B \subseteq \Omega$ and $\omega \in \Omega \setminus B$; intuitively, delaying an addition cannot increase its benefit. For set function minimization problems, supermodularity, where the inequality flips, plays an analogous role, and the greedy rule achieves constant-factor approximation guarantees (Liberty & Sviridenko, 2017).

The suboptimality of the greedy algorithm has been extensively studied for both set-function minimization (Bounia & Koriche, 2023) and maximization (Das & Kempe, 2018; Feige et al., 2011; Bian et al., 2017; Harshaw et al., 2019) when standard assumptions–non-negativity, monotonicity, and sub/supermodularity–are weakened. In contrast, results on greedy sequence maximization (Streeter & Golovin, 2008; Alaei et al., 2021; Zhang et al., 2015; Bernardini et al., 2020; Van Over et al., 2024; Tschiatschek et al., 2017)–where the ordering of the elements affects the function–have led to sequential analogues of submodularity and monotonicity. We leverage these tools to analyze the suboptimality of the greedy solution to minimizing the final error.

## 3 General framework for Hybrid Solvers

To solve Equation (2), consider the following hybrid iterative update:

$$u_h^{(t+1)} = u_h^{(t)} + C_{S_t} \left( f_h - \mathcal{L}_h^a u_h^{(t)} \right) \tag{4}$$

where $\mathcal{C} = \{C_j\}_{j=1}^K$ denotes a set of $K$ preconditioning functions and $S_t \in [K] = \{1, \ldots, K\}$ indexes the function chosen at step $t$. Here, we use "preconditioning function" broadly to refer to any update rule, encompassing classical approaches, where $C_j(x) = C_j x$ is a preconditioning matrix, and learned models, such as neural operators. This update generalizes the form of classical methods, as seen in Equation (3), by allowing $C_j$ to be non-linear and enabling the preconditioning function to be adaptively chosen at every step. $\mathcal{C}$ can also accommodate parameterized solver families (e.g., different Jacobi relaxation weights or multigrid cycle depths), enabling adaptive selection of solver parameters. As the number of solvers $K$ grows, it raises the likelihood of a substantial immediate error drop. Furthermore, HINTS (Zhang et al., 2024) is a special case of this hybrid solver with $K = 2$, where $C_1$ is a neural operator and $C_2$ is a classical preconditioner. Its routing rule is given by $S_t = \mathbf{1}_{t \bmod \tau > 0} + 1$, which selects $C_1$ every $\tau$ steps and $C_2$ otherwise.

Let $e_h^{(t)} = u_h - u_h^{(t)}$ denote the error at step $t$. Then,

$$e_h^{(t+1)} = u_h - u_h^{(t)} - C_{S_t} \left( \mathcal{L}_h^a u_h - \mathcal{L}_h^a u_h^{(t)} \right) = (I - C_{S_t} \circ \mathcal{L}_h^a) \left( e_h^{(t)} \right)$$

where $I$ is the identity map and $I - C_j \circ \mathcal{L}_h^a$ is the error propagation function for the $j^{\text{th}}$ solver. Here, "$\circ$" denotes function composition (for linear $C_j$, this coincides with matrix multiplication). The objective of the hybrid solver is to select a sequence of solvers that minimize the error norm after $T$ steps or $\|e_h^{(T)}\|_2^2$. For a sequence $S = (S_1, \ldots, S_T)$ of solver indices, we seek to solve

$$\min_{S_t \in [K], |S| \leq T} h(S) := \left\| \left(I - C_{S_{|S|}} \circ \mathcal{L}_h^a\right) \circ \cdots \circ \left(I - C_{S_1} \circ \mathcal{L}_h^a\right) \left(e_h^{(0)}\right) \right\|_2^2 \tag{5}$$

with compositions applied from right to left, so that the $S_1$ update acts on the initial error.

## 4 GREEDY ALGORITHM

Equation (5) defines a combinatorial optimization problem with a search space exponential in $T$, rendering exact optimization intractable for large $T$ or $K$. As a starting point, we propose and analyze an "omniscient" greedy algorithm that assumes access to the true initial error $e_h^{(0)}$. This is unrealistic–if $e_h^{(0)}$ were known, one could recover the solution immediately via $u_h = u_h^{(0)} + e_h^{(0)}$, but it provides a clean benchmark. In Section 5, we relax this assumption using a practical learning strategy that is Bayes consistent with this omniscient approach, thereby recovering the suboptimality guarantees shown below.

As discussed in Section 2.3, when supermodularity and monotonicity hold, the greedy rule–such as the one described in Algorithm 1–enjoys constant-factor approximation guarantees. However, classical results focused on *set* functions, with only recent extensions made to sequences. Building on this line, we introduce a sequence-based notion of weak supermodularity.

---

**Algorithm 1** Greedy Algorithm for a Hybrid PDE solver

**Require:** $\{C_j\}_{j=1}^K, T, \mathcal{L}_h^a, e_h^{(0)}$
$\quad S^0 \leftarrow \emptyset$
$\quad$ **for** $t < T$ **do**
$\quad\quad S^{t+1} \leftarrow S^t \oplus \arg\min_{j \in [K]} \|(I - C_j \circ \mathcal{L}_h^a)(e_h^{(t)})\|_2^2$
$\quad\quad e_h^{(t+1)} \leftarrow (I - C_{S_{t+1}} \circ \mathcal{L}_h^a)(e_h^{(t)})$
$\quad$ **end for**
$\quad$ **return** $S^T$

---

Let $\Omega^*$ denote the space of sequences with elements in $\Omega$. For $S = (S_1, \ldots, S_n) \in \Omega^*$, $S' = (S_1', \ldots, S_m') \in \Omega^*$, we denote their concatenation as $S \oplus S' = (S_1, \ldots, S_n, S_1', \ldots, S_m')$. $S$ is a prefix of $S'$ or $S \preceq S'$ if $S' = S \oplus L$ for some $L \in \Omega^*$. A sequence function $g : \Omega^* \to \mathbb{R}$ is considered prefix monotonically non-increasing if $g(S \oplus S') \leq g(S)$ for all $S, S' \in \Omega^*$, and postfix monotonically non-increasing if $g(S' \oplus S) \leq g(S)$ for all $S, S' \in \Omega^*$. A prefix non-increasing function $g$ is sequence supermodular if, for all $S', S \in \Omega^* : S \preceq S'$, it holds that

$$g(S) - g(S \oplus \omega) \geq g(S') - g(S' \oplus \omega), \quad \forall \omega \in \Omega \tag{6}$$

However, $h(S)$ (described in Equation (5)) may not, in general, satisfy this property. Therefore, we introduce weak sequence supermodularity. A prefix non-increasing function $g$ is weakly supermodular with respect to $S' \in \Omega^*$ if, for any $S \in \Omega^{|S'|}$, there exists $\alpha(S') \geq 1$ such that

$$g(S) - g(S \oplus S') \leq \alpha(S') \sum_{i \in [|S'|]} g(S) - g(S \oplus S_i') \tag{7}$$

The parameter $\alpha(S')$ or the supermodularity ratio quantifies deviation from exact sequence supermodularity. Expanding the $g(S) - g(S \oplus S')$ as a telescoping sum $\sum_{i=1}^{|S'|} g(S \oplus (S_1', \ldots, S_{i-1}')) - g(S \oplus (S_1', \ldots, S_i'))$ shows that the marginal decrease from appending $S_i'$ after its predecessors is controlled by the effect of appending $S_i'$ directly to $S$. Thus, postponing the inclusion of $S_i'$ cannot yield a significantly larger benefit compared to adding it earlier. The supermodularity ratio $\alpha(S)$ quantifies the extent to which future gains from delays may exceed immediate gains.

Having introduced these notions, we now characterize the suboptimality of greedy solutions of weakly supermodular and postfix monotonic sequence functions in Theorem 4.1.

**Theorem 4.1.** *Let $g : \Omega^* \to \mathbb{R}$ be a weakly supermodular function with respect to the optimal solution $O = \arg\min_{S \in \Omega^T} h(S)$ with a supermodularity ratio of $\alpha(O)$ and postfix monotonicity. Let the greedy solution of length $T$ be $S^T$. If $\phi_T(\alpha) = \left(1 - \frac{1}{\alpha T}\right)^T$, then*

$$g(S^T) \leq (1 - \phi_T(\alpha(O)))\, g(O) + \phi_T(\alpha(O))g(\emptyset)$$

The proof of Theorem 4.1 appears in Appendix B.2. As $T \to \infty$, the factor $1 - \phi_T(\alpha(O))$ decreases to $1 - e^{-1/\alpha(O)}$, so the worst-case performance of the greedy rule saturates with horizon rather than degrading indefinitely. Additionally, larger $\alpha(O)$, which indicates higher reward for delayed inclusions, loosens the suboptimality bound. Theorem 4.1 requires that the sequence objective $h$ be weakly supermodular with respect to the optimal solution and postfix monotone–properties established in Proposition 4.2. We defer the proof to Appendix B.3.

**Proposition 4.2.** *Suppose that for all $j \in [K]$, the error propagation function $I - C_j \circ \mathcal{L}_h^a$ is $\rho_j$-Lipschitz continuous with $\rho_j < 1$, and that $(I - C_j \circ \mathcal{L}_h^a)(0_N) = 0_N$. Then, the function $h$ is weakly supermodular with respect to the optimal solution $O$, with*

$$\alpha(O) = \max \left\{ \frac{4}{T - \sum_{i=1}^{T} \rho_{O_i}^2}, 1 \right\}$$

*Furthermore, if $I - C_j \circ \mathcal{L}_h^a$ is invertible for all $j \in [K]$, $h$ is also postfix non-increasing.*

The conditions of Proposition 4.2 are quite natural. For classical solvers, the Lipschitz constant is $\|I_N - C_j \mathcal{L}_h^a\|$, which is usually less than 1 for well-posed linear elliptic PDEs (i.e., the update is damping errors). For neural networks, Lipschitz continuity can be enforced via weight regularization (Gouk et al., 2021) and a sufficiently trained model should approximate $(\mathcal{L}_h^a)^{-1}$ well enough to make the Lipschitz constant small. The requirement $(I - C_j \circ \mathcal{L}_h^a)(0_N) = 0_N$ is both natural and desirable: it precludes spurious updates when the residual $f_h - \mathcal{L}_h^a u_h^{(t)}$ is 0. This holds by design for classical schemes, and it can be enforced for any learned model by excluding bias terms.

The form of $\alpha(O)$ in Proposition 4.2 highlights that the suboptimality factor in Theorem 4.1 is governed by the collective contraction factors of the solvers chosen by the optimal solution: as $\sum_{i=1}^{T} \rho_{O_i}^2 \to T$, $\alpha(O)$ grows, resulting in a weaker bound. Invertibility of the error propagation functions is often satisfied with Jacobi and Gauss-Seidel updates. However, widely-used solvers like two-grid corrections and neural networks with dimension changes or ReLU activations may yield non-invertible error propagation functions. Nevertheless, our experiments show that greedy routers remain effective even when invertibility is not met.

Theorem 4.1 thus indicates that the approximation guarantee is strongest when the sequence is nearly supermodular ($\alpha(O) \approx 1$). Generally, if all error propagation maps share an eigenbasis, supermodularity is established. This occurs, for example, for linear, constant-coefficient PDEs with periodic boundary conditions where the solver ensemble includes Jacobi, Gauss-Seidel, and a single-layer linear Fourier Neural Operator. The proof of Proposition 4.3 is deferred to Appendix B.5.

**Proposition 4.3.** *Let $\|I_N - C_j \mathcal{L}_h^a\| \leq 1$ for all $j \in [K]$ and $(I - C_j \mathcal{L}_h^a) = P\Lambda_j P^{-1}$. Then, $h$ is supermodular.*

## 5 APPROXIMATE GREEDY ROUTER

The results in Section 4 indicate that the error reduction from the greedy solution closely matches that of the optimal sequence, but this is predicated on having an accurate estimate of the initial error, $e_h^{(0)}$. A poor approximation of $e_h^{(0)}$ can result in miscalibrated decisions, causing errors to amplify over subsequent steps. To remedy this, we design a router $r$ that learns to select solvers myopically, as described in Algorithm 1, without access to the true initial error.

We adopt the following learning setup. Let $\mathcal{A}, \mathcal{F}, \mathcal{U} \subseteq \mathbb{R}^N$ denote the spaces of coefficient, forcing, and solution functions on the grid $G_h(D)$. We assume an application-specific data distribution $\mathcal{P}_{\mathcal{A} \times \mathcal{F}}$ over $\mathcal{A} \times \mathcal{F}$ that reflects test time conditions. During training, $(a_h, f_h)$ is drawn from $\mathcal{P}_{\mathcal{A} \times \mathcal{F}}$, and a high-accuracy reference solution $u_h$ is computed, providing the true per-step error. The router $r$ is learned offline on this data; at test time, it operates without access to true errors.

At each time step $t$, $r$ selects a solver using the coefficient $a \in \mathcal{A}$, forcing $f \in \mathcal{F}$, and the current iterate $u^{(t)} \in \mathcal{U}$. If $r(a_h, f_h, u_h^{(t)}) = j$, the next iterate is computed with solver $j$, resulting in an error of $\|(I - C_j \circ \mathcal{L}_h^a)(e_h^{(t)})\|_2^2$. Learning such a router requires minimizing the following loss:

$$l_{\text{route}}\left(r, a_h, f_h, u_h^{(t)}, u_h\right) = \sum_{j=1}^K \left\|(I - C_j \circ \mathcal{L}_h^a)\left(u_h - u_h^{(t)}\right)\right\|_2^2 \mathbf{1}_{r(a_h, f_h, u_h^{(t)})=j} \tag{8}$$

The corresponding $l_{\text{route}}$-risk is defined as $\mathcal{R}_{\text{route}}(r) = \mathbb{E}_{a_h, f_h \sim \mathcal{P}_{\mathcal{A} \times \mathcal{F}}}[l_{\text{route}}(r, a_h, f_h, u_h^{(t)}, u_h)]$. For analysis, we fix the iterate generation via a teacher-forcing (Williams & Zipser, 1989; Lamb et al., 2016): during training, the iterates fed to the router are produced by an oracle greedy rollout, not by the router's own past choices. Formally, with $u_h^{(0)} = 0_N$, for $t \geq 1$,

$$j_t^* \in \operatorname{argmin}_{j \in [K]} \left\|(I - C_j \circ \mathcal{L}_h^a)\left(e_h^{(t-1)}\right)\right\|_2^2, \quad u_h^{(t)} = u_h^{(t-1)} + C_{j_t^*}\left(f_h - \mathcal{L}_h^a u_h^{(t-1)}\right)$$

Thus, $\{u_h^{(t)}\}$ is a deterministic function of $(a_h, f_h)$. At each $t$, $l_{\text{route}}$ is evaluated on $r(a_h, f_h, u_h^{(t)})$ which takes in the teacher-forced iterate. Finally, the greedy choice $j_{t+1}^*$ is used to advance $u_h^{(t+1)}$. Thus the input distribution seen by $r$ depends on $(a_h, f_h)$, not on the router's predictions. However, full teacher forcing induces a distributional mismatch at test time (exposure bias). In experiments, we mitigate this with scheduled sampling (Bengio et al., 2015); see Appendix D for details.

Minimizing $\mathcal{R}_{\text{route}}(r)$ yields the following Bayes-Optimal Router:

$$r^*(a_h, f_h, u_h^{(t)}) \in \operatorname{argmin}_{j \in [K]} \left\|(I - C_j \circ \mathcal{L}_h^a)\left(e_h^{(t)}\right)\right\|_2^2 \tag{9}$$

which aligns with the decision rule in Algorithm 1. Hence, $l_{\text{route}}$ is consistent with learning the greedy algorithm.

## 5.1 SURROGATE LOSS

Since Equation (8) is discontinuous and non-convex, direct minimization is intractable in practice. To overcome this, we introduce a surrogate loss that satisfies three key properties: (1) convexity, enabling efficient optimization; (2) serving as an upper bound on the original loss; and (3) Bayes consistency, ensuring the Bayes optimal decision of the original loss is preserved upon minimization. Formally, a surrogate $\phi$ is considered to be Bayes consistent with respect to the loss $l$ if

$$\lim_{n \to \infty} \mathcal{R}_\phi(f_n) - \mathcal{R}_\phi^* \implies \lim_{n \to \infty} \mathcal{R}_l(f_n) - \mathcal{R}_l^*$$

where $\mathcal{R}_l(f) = \mathbb{E}[l(f(X), Y)]$ and $\mathcal{R}_l^* = \inf_f \mathcal{R}_l(f)$. In other words, in the limit of infinite data, if the risk of a sequence of learned hypotheses $\{f_n\}$ converges to the optimal risk under $\phi$, it also converges to the optimal risk with respect to the original loss $l$.

To define a surrogate loss for the routing problem, consider a set of scoring functions $\mathbf{g} = \{g_j\}_{j=1}^K$ with $g_j : \mathcal{A} \times \mathcal{F} \times \mathcal{U} \to \mathbb{R}$ and we define the router as $r(a, f, u^{(t)}) = \operatorname{argmax}_{j \in [K]} g_j(a, f, u^{(t)})$. For example, $\mathbf{g}$ can be a neural network with $K$ outputs. Then, minimizing the following surrogate loss yields the same decision as minimizing Equation (8):

$$\Psi\left(\mathbf{g}, a_h, f_h, u_h^{(t)}, u_h\right) = -\sum_{j=1}^K \sum_{k=1}^K \tilde{c}_k(a_h, u_h^{(t)}, u_h) \mathbf{1}_{k \neq j} \log\left(\frac{\exp\left(g_j(a, f, u^{(t)})\right)}{\sum_{m=1}^K \exp\left(g_m(a, f, u^{(t)})\right)}\right) \tag{10}$$

where $\tilde{c}_j(a_h, u_h^{(t)}, u_h) = \|(I - C_j \circ \mathcal{L}_h^a)(u_h - u_h^{(t)})\|_2^2$. The $\Psi$-risk is denoted by $\mathcal{R}_\Psi(\mathbf{g}) = \mathbb{E}_{a_h, f_h \sim \mathcal{P}_{\mathcal{A} \times \mathcal{F}}}[\Psi(\mathbf{g}, a_h, f_h, u_h^{(t)}, u_h)]$. The convexity of $\Psi$ with respect to $\mathbf{g}$ follows from the convexity of log-softmax function in its inputs. Moreover, $\Psi$ upper bounds $l_{\text{route}}$ up to a constant factor, and we refer the reader to Appendix C.2 for the proof. Finally, Theorem 5.1 shows that $\Psi$ achieves Bayes consistency with respect to $l_{\text{route}}$; the proof can be found in Appendix C.3.

**Theorem 5.1.** *Let* $\tilde{c}_j(a_h, u_h^{(t)}, u_h) < \bar{E} < \infty$ *for all* $j \in [K]$. *If there exists* $j \in [K]$ *such that* $\tilde{c}_j(a_h, u_h^{(t)}, u_h) > E_{min} > 0$, *then, for any collection of solvers* $\{C_j\}_{j=1}^K$ *and linear discrete operator* $\mathcal{L}_h^a$, $\Psi$ *is Bayes consistent surrogate for* $l_{route}$.

Theorem 5.1 is a cost-sensitive analogue of the classical Bayes-consistency of multiclass cross-entropy for 0-1 loss: in the infinite-sample limit, minimizing cross-entropy recovers the true conditional class probabilities, so the induced decision is Bayes optimal. Our result extends this to cross-entropy with instance-dependent weights $\sum_{k \neq j}^{K} \tilde{c}_k(a_h, u_h^{(t)}, u_h)$. The uniform upper bound holds when all preconditioning functions are error-damping. It is also reasonable to assume at least one solver cannot annihilate the error in one step, yielding the lower bound. Under these conditions, minimizing $\Psi$ recovers the Bayes-optimal router of Equation (9), i.e., the greedy solution of Equation (5). Following the work of Mao et al. (2024), the proof uses standard conditional-risk calibration: we relate the excess risks of $l_{\text{route}}$ and $\Psi$ and take the infinite-sample limit.

## 6 RELATED WORKS

**Hybrid PDE Solvers:** Early data-driven solvers sought convergence guarantees by predicting parameters–e.g., preconditioning matrices $C$, multi-grid smoothers or restriction matrices–within iterative schemes (Taghibakhshi et al., 2021; Caldana et al., 2024; Kopaničáková & Karniadakis, 2025; Huang et al., 2022; Katrutsa et al., 2020). These works, however, do not leverage the neural surrogates' ability to generalize across varying coefficients and/or forcings highlighted in Section 2.2 and thus, offer only modest speedups over classical solvers. HINTS (Zhang et al., 2024) introduced hybrid solvers that interleave a classical method with a pre-trained DeepONet on a fixed schedule (e.g., 24 Jacobi steps, then one DeepONet correction). Despite this rigidity, HINTS reports significantly faster convergence than the standalone numerical solver. Kahana et al. (2023) adapted HINTS to geometries distinct from, but related to, those used to train the neural operator. Both Hu & Jin (2025) and Cui et al. (2022) characterized the dampening of error modes over iterates, with the former replacing the DeepONet of HINTS with an MIONet (Jin et al., 2022) and the latter a Fourier Neural Operator (Li et al., 2020a). Critically, these hybrid solvers are limited in two ways: they, firstly, only complement the trained surrogate with a *single* numerical solver, and secondly interleave the numerical and neural solvers with a fixed, heuristic schedule. These were the primary shortcomings that we addressed in our proposed method, which result in significant empirical improvements, as we demonstrate in Section 7.

**Model Routing:** Routing (Shnitzer et al., 2023; Hu et al., 2024; Ding et al., 2024; Huang et al., 2025) seeks to find, for each input, a model from a fixed set that optimizes a task metric under cost or latency constraints. Simple heuristics–e.g., thresholding a cheap model's uncertainty estimates (Chuang et al., 2024; 2025)–often poorly balance the cost-accuracy tradeoff. Consequently, many systems learn a router network, which maps a query to the model index expected to perform best (Hari & Thomson, 2023; Mohammadshahi et al., 2024; Šakota et al., 2024). Abstractly, our method also learns a router, but over numerical and neural solvers for PDEs at the iteration level. Our work is the first to demonstrate that routing is applicable to the problem of learning hybrid PDE solvers, differing from prior hybrid solver works, which simply fixed the "router" to a predetermined schedule. We additionally exploit the algebraic structure of PDE solvers to derive theoretical guarantees for our routing strategy in Section 4, unlike typical model-routing settings.

**Mixture of Experts:** Classical mixture-of-experts (MoE) (Jacobs et al., 1991; Jordan & Jacobs, 1994) uses a gating network to assign soft or hard weights to a set of experts and produce a weighted combination of their outputs, with the gate and experts trained jointly. In modern LLM systems, MoE instead performs sparse, token-level routing to a small subset of in-layer experts, enabling large model capacity without proportional compute, typically with load-balancing and capacity constraints (Shazeer et al., 2017; Lepikhin et al., 2020; Fedus et al., 2022; Jiang et al., 2024). Our method can be regarded as gating network of a different kind where our "experts" or solvers are not trained jointly with the router and single router is reused across the iteration horizon, in contrast to MoE which commonly employs layer-specific gates.

## 7 EXPERIMENTS

In this section, we empirically demonstrate the fast, uniform convergence of the approximate greedy router on Poisson and Helmholtz equations posed on the unit domain $D = [0, 1]^d$ with $d \in \{1, 2\}$.

| Equation | 1D Poisson | | 2D Poisson | | 1D Helmholtz | | 2D Helmholtz | |
|---|---|---|---|---|---|---|---|---|
| Methods | $\|e_h^{(T)}\|$ | AUC | $\|e_h^{(T)}\|$ | AUC | $\|e_h^{(T)}\|$ | AUC | $\|e_h^{(T)}\|$ | AUC |
| Jacobi-related Solvers | | | | | | | | |
| Jacobi Only | 0.19 (0.1) | 128.66 (69.98) | 0.07 (0.03) | 115.5 (42.96) | 0.22 (0.11) | 146.57 (76.11) | **0.066 (0.03)** | **110.485 (41)** |
| HINTS-Jacobi | 0.01 | 22.71 (11.36) | 0.18 | 65.75 (8.61) | 0.048 (0.02) | 29.36 (13.21) | 0.11 (0.03) | 115.84 (41.36) |
| Greedy-Jacobi | **0.001** | **1.12 (0.44)** | **0.01** | **18.48 (4.06)** | **0.025 (0.03)** | **8.105 (8.22)** | **0.066 (0.03)** | **110.485 (41)** |
| GS-related Solvers | | | | | | | | |
| GS Only | 0.05 (0.03) | 80.55 (43.83) | 0.005 | 61.67 (23.02) | 0.054 (0.03) | 91.153 (47.35) | **0.005 (0.002)** | **58.801 (21.892)** |
| HINTS-GS | 0.003 | 17.8 (9.21) | 0.174 | 56.63 (7.75) | 0.041 (0.01) | 26.638 (11.59) | 0.056 (0.003) | 64.445 (21.986) |
| Greedy-GS | $< 10^{-3}$ | **0.674 (0.27)** | **0.001** | **9.9 (2.26)** | **0.015 (0.01)** | **5.157 (3.96)** | **0.005 (0.002)** | **58.801 (21.892)** |
| MG-related Solvers | | | | | | | | |
| MG Only | 0.05 (0.01) | 20.51 (7.46) | 0.002 | 10.97 (3.03) | 0.05 (0.01) | 22.042 (7.8) | **0.011 (0.009)** | **10.745 (3.094)** |
| HINTS-MG | 0.002 | 7.64 (3.66) | 0.079 | 15.02 (2.03) | 0.023 (0.01) | 9.872 (4.32) | 0.026 (0.008) | 11.763 (2.976) |
| Greedy-MG | $< 10^{-3}$ | **0.24 (0.1)** | **0.001** | **2.58 (0.54)** | **0.016 (0.01)** | **1.784 (1.32)** | **0.011 (0.009)** | **10.745 (3.094)** |

Table 1: Final error and AUC of squared $L^2$ error (lower is better). Values are mean ($\pm$ standard error (s.e.)) over 64 test instances; both mean and s.e. are reported in $\times 10^{-3}$. If a standard error is not shown, it is $< 10^{-3}$ in the reported units (raw $< 10^{-6}$). Bold indicates the best method within each solver family.

For Poisson, we use the constant-coefficient model

$$-\Delta u(x) = f(x), \quad x \in D$$

and for Helmholtz, we adopt the sign convention

$$-\Delta u(x) - a^2(x)u(x) = f(x), \quad x \in D$$

where $a^2(x) \geq 0$. We impose periodic boundary conditions for both equations: for Poisson, the right-hand side is centered to satisfy the compatibility condition $\int_D f(x)dx = 0$. The domain $D$ is discretized on uniform grids with 65 points in 1D and a $33 \times 33$ grid in 2D. Data is sampled from a zero-mean Gaussian Random Field on the periodic domain with covariance operator $(-\Delta + 9I)^{-2}$. We run two experiments: (i) compare our greedy method to HINTS and single-solver baselines, and (ii) assess how performance scales as the solver ensemble grows. For both experiments, the performance across 64 test samples is reported via the mean final error $\|e_h^{(T)}\|_2$ and the mean area under the curve (AUC), where AUC $= \sum_{t=1}^{T} \|e_h^{(t)}\|_2$. While the final errors is of utmost importance, AUC captures performance over the entire run–smaller values indicate lower error at all intermediate steps–and naturally penalizes non-monotone spikes that undo progress.

**Comparing Greedy with HINTS:** For this experiment, we consider Jacobi, Gauss-Seidel (GS), and multigrid (MG) solvers along with a DeepONet model. As baselines, we use single-solver schedules (Jacobi only, GS only, and MG only) as well as HINTS variants with each classical solver (HINTS-Jacobi, HINTS-GS, HINTS-MG), where the DeepONet correction is interleaved every 24 Jacobi/GS iteration or every 14 MG V-cycles. We then train LSTM-based routers using the loss in Equation (10) under three different solver-access sets: Jacobi+DeepONet (Greedy-Jacobi), GS+DeepONet (Greedy-GS), and MG+DeepONet (Greedy-MG). We use an LSTM because solver routing is sequential and the benefit of a greedy step depends on the trajectory of errors and past choices. LSTMs have historically performed well on sequence data owing to their recurrent memory. Training details of the DeepONet and the routers can be found in Appendix D.

We note that comparisons are restricted to methods with the same solver access. For example, Greedy-Jacobi and HINTS-GS are not comparable because the former lacks access to GS. Accordingly, we partition results by solver family (Jacobi-related, GS-related, and MG-related). Each method is run for 300 iterations (Jacobi/GS) or 100 V-cycles (MG).

As shown in Table 1, across all solver families, the greedy router achieves the lowest final error and lowest mean AUC: Greedy-Jacobi, Greedy-GS, and Greedy-MG outperform both their single-solver and HINTS counterparts. While HINTS improves over single-solver schedules, it exhibits sawtooth error traces (See Figure 1) as HINTS often invokes the DeepONet when it is suboptimal. By contrast, the greedy routers only routes to solvers that yield an immediate error drop, which translates into the reported AUC gains and near-monotone error decay. Notably, the greedy solutions for 2D Helmholtz largely follow the corresponding classical solver alone. This indicates that the learned neural correction frequently fails to reduce the error and is therefore skipped by the greedy rule. HINTS, however, continues to call the neural operator at a fixed interval even when it increases the error, which explains its substantially worse AUC and final error on 2D Helmholtz.

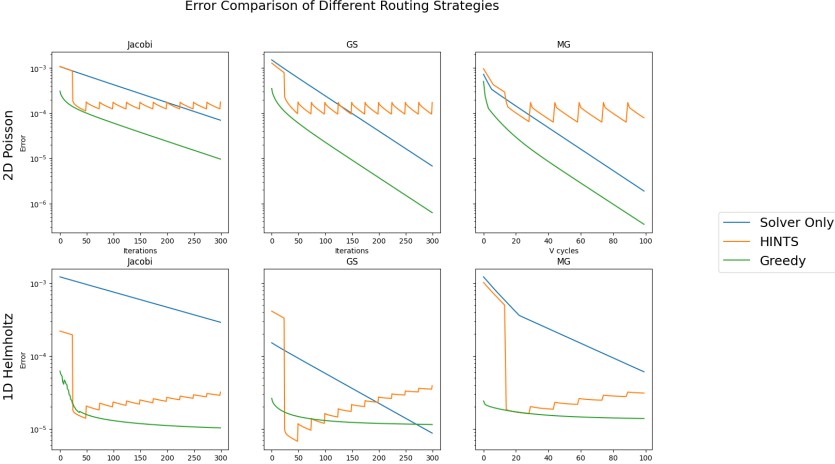

Figure 1: Convergence histories for representative test instances. Rows: 2D Poisson (top) and 1D Helmholtz (bottom). Columns: Jacobi, Gauss–Seidel (GS), and multigrid (MG). Greedy yields near-monotone decay and the lowest errors, whereas HINTS shows sawtooth behaviors. Convergence histories for 1D Poisson and 2D Helmholtz to Appendix E

| Equation | 1D Poisson | | 1D Helmholtz | |
|---|---|---|---|---|
| # of Solvers | $\|e_h^{(T)}\|$ | AUC | $\|e_h^{(T)}\|$ | AUC |
| 2 | 0.002 (0.001) | 1.717 (0.643) | 0.018 (0.013) | 6.665 (3.81) |
| 3 | 0.002 (0.001) | 1.434 (0.554) | 0.017 (0.014) | 6.31 (3.852) |
| 4 | 0.002 (0.001) | 1.337 (0.523) | 0.017 (0.014) | 6.182 (3.868) |
| 5 | 0.001 (0.001) | 1.293 (0.508) | 0.017 (0.014) | 6.113 (3.878) |
| 6 | 0.001 (0.001) | 1.121 (0.449) | 0.017 (0.014) | 6.098 (3.88) |

Table 2: Final error and AUC of squared $L^2$ error for varying numbers of solvers. Values are mean ($\pm$ standard error (s.e.)) over 64 test instances; both mean and s.e. are reported in $\times 10^{-3}$.

**Size of solver ensembles:** We also study how the size of the solver ensemble affects final error and AUC. Each router is trained with an ensemble that always includes a DeepONet and a subset of weighted Jacobi solvers with relaxation parameters $\omega \in \{0.5, 0.67, 0.75, 0.8, 1\}$. We grow the solver set cumulatively, starting from the smallest $\omega$ and adding larger ones. In Table 2, the AUC decreases as the size of the solver set increases for both 1D Poisson and 1D Helmholtz (300 iterations), but with diminishing returns.

Additional experiments are provided in Appendix E.

## 8 DISCUSSION

We have introduced an adaptive method for selecting solvers that efficiently minimize the final error when solving a PDE iteratively. This opens several directions for future work. Our current DeepONet is trained in isolation, without accounting for its downstream role as a correction term predictor. Jointly training the ML solver and the router could yield larger gains. Another avenue is to employ reinforcement learning to learn a cost-aware routing strategy that optimizes the terminal error under compute budgets. When the deployment conditions are unknown or volatile, our offline training procedure suffers and a fixed schedule like HINTS can perform better, however, framing routing as an online learning problem enables continual adaptation. Finally, it would be interesting to extend this routing scheme to broader optimization settings in which the router selects from a suite of optimizers at each iteration.

## REPRODUCIBILITY STATEMENT

The implementation of our hybrid PDE solver with learned greedy routing is provided in the source code submitted as supplemental materials. It includes scripts to reproduce all tables and figures in Section 7 and Appendix E, along with a README file that documents dependencies and commands for training routers and instructions for regenerating results. Data is generated on the fly using fixed seeds (specified in the code) to ensure exact reproducibility. Formal proofs of all theorems and propositions appear in Appendices B and C.

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

## A  BACKGROUND

### A.1  MULTIGRID

Let $A_h$, $A_{2h}$ represent the coefficient matrix on a fine grid with discretization parameter $h$ and $2h$, $u_h$ and $f_h$ represent the PDE solution and constant vector on a grid discretized by $h$, $R_h^{2h}$ denote the restriction matrix which transfers vectors from a fine grid to a coarse one, and $I_{2h}^h$ is the interpolation matrix transfers vectors from a coarse grid to a fine one. In a 2-grid method, a few iterations of the smoother (e.x. Jacobi or Gauss-Seidel) are first applied on the fine grid to approximate the solution of $A_h u_h = f_h$. The residual is then computed as $r_h = f_h - A_h u_h$ and restricted to the coarse grid via $r_{2h} = R_h^{2h} r_h$. The error equation, $A_{2h} e_{2h} = r_{2h}$, is solved on the coarse grid. The resulting estimate of the error is interpolated in the fine grid by $e_h = I_{2h}^h e_{2h}$, and the fine grid solution is updated by adding this correction, $u_h = u_h + e_h$. Finally, additional smoothing steps are performed on the fine grid to further reduce any high frequency errors. The preconditioning matrix for two-grid solver is $C_{2G} = I_{2h}^h A_{2h}^{-1} R_h^{2h}$. More complex strategies for multigrid like V-cycle and W-cycle compute error corrections recursively across multiple grids of varying coarseness

## B  PROOFS FOR SECTION 4

### B.1  PROOF OF PROPOSITION B.1

**Proposition B.1.** *Any prefix monotonically non-increasing sequence supermodular function $g$ is weakly supermodular with respect to all sequences $S \in \Omega^*$ with $\alpha(S) = 1$*

*Proof.* This proof is adapted from Liberty & Sviridenko (2017).

If $g$ is sequence supermodular then,

$$g(S) - g(S \oplus S')$$

$$= \sum_{i=1}^{|S'|} g(S \oplus (S'_1, \ldots, S'_{i-1})) - g(S \oplus (S'_1, \ldots, S'_i))$$

$$\overset{(a)}{\le} \sum_{i=1}^{|S'|} g(S) - g(S \oplus S'_i)$$

$$\le |S'| \max_{i \in [|S'|]} g(S) - g(S \oplus S'_i)$$

(a) by supermodularity

$\square$

## B.2 Proof of Theorem 4.1

**Theorem 4.1.** *Let $g : \Omega^* \to \mathbb{R}$ be a weakly supermodular function with respect to the optimal solution $O = \arg\min_{S \in \Omega^T} h(S)$ with a supermodularity ratio of $\alpha(O)$ and postfix monotonicity. Let the greedy solution of length $T$ be $S^T$. If $\phi_T(\alpha) = \left(1 - \frac{1}{\alpha T}\right)^T$, then*

$$g(S^T) \leq (1 - \phi_T(\alpha(O)))\, g(O) + \phi_T(\alpha(O)) g(\emptyset)$$

*Proof.* This proof strategy is inspired by Streeter & Golovin (2008) and Liberty & Sviridenko (2017)

$$
\begin{aligned}
g(S^t) - g(O) &\overset{(a)}{\leq} g(S^t) - g(S^t \oplus O) \\
&= \sum_{i=1}^{|O|} g(S^t \oplus (o_1, \ldots o_{i-1})) - g(S^t \oplus (o_1, \ldots, o_i)) \\
&\overset{(b)}{\leq} \alpha(O) \sum_{i=1}^{|O|} g(S^t) - g(S^t \oplus o_i) \\
&\leq \alpha(O)|O| \max_{i \in [|O|]} g(S^t) - g(S^t \oplus o_i) \\
&\leq \alpha(O)T g(S^t) - \alpha(O)T \min_{\omega \in \Omega} g(S^t \oplus \omega) \\
&= \alpha(O)T g(S^t) - \alpha(O)T g(S^{t+1})
\end{aligned}
$$

(a) by $\mu$- postfix monotonicity, (b) by supermodularity.

After rearranging the inequality, we get:

$$
\begin{aligned}
g(S^{t+1}) &\leq \frac{1}{\alpha(O)T} \left(g(O) - (\alpha(O)T - 1)\, g(S^t)\right) \\
&= \frac{1}{\alpha(O)T} g(O) + \left(1 - \frac{1}{\alpha(O)T}\right) g(S^t)
\end{aligned}
$$

When recursively applying this inequality, we get:

$$
\begin{aligned}
g(S^T) &\leq \frac{1}{\alpha(O)T} g(O) \sum_{i=0}^{T-1} \left(1 - \frac{1}{\alpha(O)T}\right)^i + \left(1 - \frac{1}{\alpha(O)T}\right)^T g(\emptyset) \\
&= \left(1 - \left(1 - \frac{1}{\alpha(O)T}\right)^T\right) g(O) + \left(1 - \frac{1}{\alpha(O)T}\right)^T g(\emptyset)
\end{aligned}
$$

$\square$

## B.3 Proof of Proposition 4.2

**Proposition 4.2.** *Suppose that for all $j \in [K]$, the error propagation function $I - C_j \circ \mathcal{L}_h^a$ is $\rho_j$-Lipschitz continuous with $\rho_j < 1$, and that $(I - C_j \circ \mathcal{L}_h^a)(0_N) = 0_N$. Then, the function $h$ is weakly supermodular with respect to the optimal solution $O$, with*

$$\alpha(O) = \max\left\{ \frac{4}{T - \sum_{i=1}^{T} \rho_{O_i}^2}, 1 \right\}$$

*Furthermore, if $I - C_j \circ \mathcal{L}_h^a$ is invertible for all $j \in [K]$, $h$ is also postfix monotonically non-increasing.*

*Proof.* For brevity, we use the notation $(g_1 \circ \cdots \circ g_T)(x) = \circ_{t=1}^T g_t(x)$, where composition is applied from right to left so that $g_T$ acts first. In this proof, we use a few properties of Lipschitz continuous functions:

- **Property 1:** If $g$ is $\rho$-Lipschitz continuous and $g(0) = 0$, then $\|g(x)\|_2 = \|g(x) - 0\|_2 = \|g(x) - g(0)\|_2 \le \rho\|x - 0\|_2 = \rho\|x\|_2$

- **Property 2:** If $g_1$ and $g_2$ are Lipschitz continuous functions with Lipschitz constants of $\rho_1$ and $\rho_2$ respectively, the Lipschitz constant of $g_1 + g_2$ and $g_1 - g_2$ is $\rho_1 + \rho_2$.

- **Property 3:** If $g_1$ and $g_2$ are Lipschitz continuous functions with Lipschitz constants of $\rho_1$ and $\rho_2$ respectively, the Lipschitz constant of $g_1 \circ g_2$ is $\rho_1 \rho_2$.

In order to prove weakly-$\alpha$-supermodularity, we must first prove prefix monotonicity.

**Prefix monotonicity:** Let $S \preceq S'$ where $S' = S \oplus N$.

$$h(S') = \left\| \circ_{t=|S'|}^1 \left(I - C_{S'_t} \circ \mathcal{L}_h^a\right)\left(e_h^{(0)}\right) \right\|_2^2$$

$$= \left\| \circ_{t=|S'|}^{|S|+1} \left(I - C_{S'_t} \circ \mathcal{L}_h^a\right) \circ \circ_{t=|S|}^1 \left(I - C_{S'_t} \circ \mathcal{L}_h^a\right)\left(e_h^{(0)}\right) \right\|_2^2$$

$$\overset{(a)}{\le} \left( \prod_{t=|S|+1}^{|S'|} \rho_{S'_t}^2 \right) \left\| \circ_{t=|S|}^1 \left(I - C_{S_t} \circ \mathcal{L}_h^a\right)\left(e_h^{(0)}\right) \right\|_2^2$$

$$\le h(S)$$

(a) by Property 1

**Weak supermodularity:** We will upper bound $\alpha(O)$ by providing an upper bound for $h(S) - h(S \oplus O)$ and a lower bound for $\sum_{i=1}^T h(S) - h(S \oplus O_i)$.

$$h(S) - h(S \oplus O) = \left\| \circ_{t=|S|}^1 \left(I - C_{S_t} \circ \mathcal{L}_h^a\right)\left(e_h^{(0)}\right) \right\|_2^2 - \left\| \circ_{t=|O|}^1 \left(I - C_{O_t} \circ \mathcal{L}_h^a\right) \circ \circ_{t=|S|}^1 \left(I - C_{S_t} \circ \mathcal{L}_h^a\right)\left(e_h^{(0)}\right) \right\|_2^2$$

$$\overset{(a)}{\le} \left\| \circ_{t=|S|}^1 \left(I - C_{S_t} \circ \mathcal{L}_h^a\right)\left(e_h^{(0)}\right) - \circ_{t=|O|}^1 \left(I - C_{O_t} \circ \mathcal{L}_h^a\right) \circ \circ_{t=|S|}^1 \left(I - C_{S_t} \circ \mathcal{L}_h^a\right)\left(e_h^{(0)}\right) \right\|_2^2$$

$$= \left\| \left(I - \circ_{t=|O|}^1 \left(I - C_{O_t} \circ \mathcal{L}_h^a\right)\right) \circ \circ_{t=|S|}^1 \left(I - C_{S_t} \circ \mathcal{L}_h^a\right)\left(e_h^{(0)}\right) \right\|_2^2$$

$$\overset{(b)}{\le} \left(1 + \prod_{t=1}^{|O|} \rho_{O_t}\right)^2 \left\| \circ_{t=|S|}^1 \left(I - C_{S_t} \circ \mathcal{L}_h^a\right)\left(e_h^{(0)}\right) \right\|_2^2$$

$$\overset{(c)}{\le} 4 \left\| \circ_{t=|S|}^1 \left(I - C_{S_t} \circ \mathcal{L}_h^a\right)\left(e_h^{(0)}\right) \right\|_2^2$$

(a) by reverse triangle property and $h(S) - h(S \oplus S') > 0$ by prefix monotonicity, (b) since the Lipschitz constant of $I - \circ_{t=|S'|}^1 \left(I - C_{S'_t} \circ \mathcal{L}_h^a\right)$ is $1 + \prod_{t=1}^{|S'|} \rho_{S_t}$ by Property 2 and 3, (c) since $\rho_j < 1$

To lower bound $\sum_{i=1}^{|O|} h(S) - h(S \oplus O_i)$

$$\sum_{i=1}^{|O|} h(S) - h(S \oplus O_i) = \sum_{i=1}^{|O|} \left\| \circ_{t=|S|}^1 \left(I - C_{S_t} \circ \mathcal{L}_h^a\right)\left(e_h^{(0)}\right) \right\|_2^2 - \left\| \left(I - C_{O_i} \circ \mathcal{L}_h^a\right) \circ \circ_{t=|S|}^1 \left(I - C_{S_t} \circ \mathcal{L}_h^a\right)\left(e_h^{(0)}\right) \right\|_2^2$$

$$\ge \sum_{i=1}^{|O|} \left\| \circ_{t=|S|}^1 \left(I - C_{S_t} \circ \mathcal{L}_h^a\right)\left(e_h^{(0)}\right) \right\|_2^2 - \rho_{O_i}^2 \left\| \circ_{t=|S|}^1 \left(I - C_{S_t} \circ \mathcal{L}_h^a\right)\left(e_h^{(0)}\right) \right\|_2^2$$

$$= \left\| \circ_{t=|S|}^1 \left(I_N - C_{S_t} \circ \mathcal{L}_h^a\right)\left(e_h^{(0)}\right) \right\|_2^2 \left(T - \sum_{i=1}^T \rho_{O_i}^2\right)$$

Finally,

$$\frac{h(S) - h(S \oplus O)}{T \max_i h(S) - h(S \oplus O_i)} \leq \max \left\{ \frac{4 \left\| \circ_{t=|S|}^{1} (I - C_{S_t} \circ \mathcal{L}_h^a) \left(e_h^{(0)}\right) \right\|_2^2}{\left\| \circ_{t=|S|}^{1} (I_N - C_{S_t} \circ \mathcal{L}_h^a) \left(e_h^{(0)}\right) \right\|_2^2 \left(T - \sum_{i=1}^{T} \rho_{O_i}^2\right)}, 1 \right\}$$

$$= \max \left\{ \frac{4}{T - \sum_{i=1}^{T} \rho_{O_i}^2}, 1 \right\}$$

**Postfix monotonicity:** Let $S' = S \oplus N$.

$$h(S') = \left\| \circ_{t=|S'|}^{1} \left(I - C_{S_t'} \circ \mathcal{L}_h\right) e_h^{(0)} \right\|_2^2$$

$$= \left\| \circ_{t=|N|}^{1} (I - C_{N_t} \circ \mathcal{L}_h) \circ \circ_{t=|S|}^{1} (I - C_{S_t} \circ \mathcal{L}_h) e_h^{(0)} \right\|_2^2$$

$$\stackrel{(a)}{=} \left\| \circ_{t=|N|}^{1} (I - C_{N_t} \circ \mathcal{L}_h) \circ \circ_{t=|S|}^{1} (I - C_{S_t} \circ \mathcal{L}_h) \circ \left( \circ_{t=|N|}^{1} (I - C_{N_t} \circ \mathcal{L}_h) \right)^{-1} \circ \circ_{t=|N|}^{1} (I - C_{N_t} \circ \mathcal{L}_h) e_h^{(0)} \right\|_2^2$$

$$\leq \prod_{t=1}^{|N|} \rho_{N_t}^2 \prod_{t=1}^{|S|} \rho_{S_t}^2 \prod_{t=1}^{|N|} \rho_{N_t}^{-2} \left\| \circ_{t=|N|}^{1} (I - C_{N_t} \circ \mathcal{L}_h) e_h^{(0)} \right\|_2^2$$

$$\leq h(N)$$

(a) due the invertibility of $I_N - C_j \mathcal{L}_h$

$\square$

### B.4 PROOF OF LEMMA B.2

**Lemma B.2.** *Let* $(I - C_j \mathcal{L}_h^a) = P \Lambda_j P^{-1}$ *where* $P$ *is an orthogonal matrix and* $\Lambda_j = \text{diag}(\lambda_{j1}, \ldots, \lambda_{jN})$. *If* $P^{-1} e_h^{(0)} = z$, *then the following equality holds:*

$$h(S) = \sum_{i=1}^{N} z_i^2 \prod_{j=1}^{K} \lambda_{ji}^{2m_j(S)} \tag{11}$$

*where* $m_j(S) = \sum_{t=1}^{|S|} \mathbf{1}_{S_t = j}$

*Proof.*

$$h(S) = \left\| \prod_{t=|S|}^{1} \left( I_N - C_{S_t} \mathcal{L}_h^a \right) e_h^{(0)} \right\|^2$$

$$= \left\| \prod_{t=|S|}^{1} \left( P\Lambda_{S_t} P^{-1} \right) e_h^{(0)} \right\|^2$$

$$= \left\| P \prod_{t=|S|}^{1} \Lambda_{S_t} P^{-1} e_h^{(0)} \right\|^2$$

$$= \left\| \prod_{t=|S|}^{1} \Lambda_{S_t} P^{-1} e_h^{(0)} \right\|^2$$

$$= \sum_{i=1}^{N} z_i^2 \prod_{t=T}^{1} \lambda_{S_t i}^2$$

$$= \sum_{i=1}^{N} z_i^2 \prod_{j=1}^{K} \lambda_{ji}^{2m_j(S)}$$

$\square$

## B.5   PROOF OF PROPOSITION 4.3

**Proposition 4.3.** *Let $\|I_N - C_j \mathcal{L}_h^a\| \leq 1$ for all $j \in [K]$ and $(I - C_j \mathcal{L}_h^a) = P\Lambda_j P^{-1}$. Then, $h$ is supermodular.*

*Proof.* Let $S \preceq S'$ where $S' = S \oplus B$. By Lemma B.2,

$$h(S) = \sum_{i=1}^{N} z_i^2 \prod_{j=1}^{K} \lambda_{ji}^{2m_j(S)}$$

where $m_j(S) = \sum_{t=1}^{|S|} \mathbf{1}_{S_t=j}$ be the number of times a sequence $S$ calls the solver $j$. Recall that $h$ is considered sequence supermodular if $\forall S', S \in \Omega^*$ such that $S \preceq S'$, it holds that

$$h(S) - h(S \oplus \omega) \geq h(S') - h(S' \oplus \omega)$$

$$h(S) - h(S \oplus \omega) = \sum_{i=1}^{N} z_i^2 \prod_{j=1}^{K} \lambda_{ji}^{2m_j(S)} - \sum_{i=1}^{N} z_i^2 \lambda_{\omega i}^2 \prod_{k=1}^{K} \lambda_{ji}^{2m_j(S)}$$

$$= \sum_{i=1}^{N} \left( 1 - \lambda_{\omega i}^2 \right) z_i^2 \prod_{j=1}^{K} \lambda_{ji}^{2m_j(S)}$$

Similarly,

$$h(S') - h(S \oplus \omega) = \sum_{i=1}^{N} \left( 1 - \lambda_{\omega i}^2 \right) z_i^2 \prod_{j=1}^{K} \lambda_{ji}^{2m_j(S')}$$

$$\overset{(a)}{=} \sum_{i=1}^{N} \left( 1 - \lambda_{\omega i}^2 \right) z_i^2 \prod_{j=1}^{K} \lambda_{ji}^{2(m_j(S)+m_j(B))}$$

$$\overset{(b)}{\leq} \sum_{i=1}^{N} \left( 1 - \lambda_{\omega i}^2 \right) z_i^2 \prod_{j=1}^{K} \lambda_{ji}^{2m_j(S)}$$

$$= h(S) - h(S \oplus \omega)$$

(a) Since $m_j(S') = \sum_{t=1}^{|S'|} \mathbf{1}_{S'_t=j} = \sum_{t=1}^{|S|} \mathbf{1}_{S'_t=j} + \sum_{t=|S|}^{|S'|} \mathbf{1}_{S'_t=j} = \sum_{t=1}^{|S|} \mathbf{1}_{S_t=k} + \sum_{t=1}^{|B|} \mathbf{1}_{B_t=j} = m_j(S) + m_j(B)$, (b) since $\rho(I_N - C_j\mathcal{L}_h^a) < 1$ and $m_j(B) \geq 0$ for all $j \in [K]$

$\square$

## C  PROOFS FOR SECTION 5

### C.1  PROOF OF LEMMA C.1

**Lemma C.1.** *For any set of preconditioning functions $\mathcal{C}$, any discrete operator $\mathcal{L}_h^a$, any router $r$, any $a_h, f_h, u_h^{(t)}, u_h \in \mathcal{A} \times \mathcal{F} \times \mathcal{U} \times \mathcal{U}$, the following equality holds true:*

$$
l_{route}\left(r, a_h, f_h, u_h^{(t)}, u_h\right) = \sum_{j=1}^{K} \sum_{k=1}^{K} \left\| (I - C_k \circ \mathcal{L}_h^a)\left(u_h - u_h^{(t)}\right) \right\|_2^2 \mathbf{1}_{k \neq j} \mathbf{1}_{r(a_h, f_h, u_h^{(t)}) \neq j}
$$

$$
- (K-2) \sum_{j=1}^{K} \left\| (I - C_j \circ \mathcal{L}_h^a)\left(u_h - u_h^{(t)}\right) \right\|_2^2
$$

*Proof.* Note that $\sum_{j=1}^{K} \mathbf{1}_{r(a_h, f_h, u_h^{(t)}) \neq j} = K - 1$

$$
l_{\text{route}}\left(r, a_h, f_h, u_h^{(t)}, u_h\right) = \sum_{j=1}^{K} \left\| (I - C_j \circ \mathcal{L}_h^a)\left(u_h - u_h^{(t)}\right) \right\|_2^2 \mathbf{1}_{r(a_h, f_h, u_h^{(t)})=j}
$$

$$
= \sum_{j=1}^{K} \left\| (I - C_j \circ \mathcal{L}_h^a)\left(u_h - u_h^{(t)}\right) \right\|_2^2 - \sum_{j=1}^{K} \left\| (I - C_j \circ \mathcal{L}_h^a)\left(u_h - u_h^{(t)}\right) \right\|_2^2 \mathbf{1}_{r(a_h, f_h, u_h^{(t)}) \neq j}
$$

$$
= \sum_{j=1}^{K} \left\| (I - C_j \circ \mathcal{L}_h^a)\left(u_h - u_h^{(t)}\right) \right\|_2^2 - \sum_{j=1}^{K} \left\| (I - C_j \circ \mathcal{L}_h^a)\left(u_h - u_h^{(t)}\right) \right\|_2^2 \mathbf{1}_{r(a_h, f_h, u_h^{(t)}) \neq j}
$$

$$
+ (K-1) \sum_{k=1}^{K} \left\| (I - C_k \circ \mathcal{L}_h^a)\left(u_h - u_h^{(t)}\right) \right\|_2^2 - (K-1) \sum_{k=1}^{K} \left\| (I - C_k \circ \mathcal{L}_h^a)\left(u_h - u_h^{(t)}\right) \right\|_2^2
$$

$$
= \sum_{j=1}^{K} \left\| (I - C_j \circ \mathcal{L}_h^a)\left(u_h - u_h^{(t)}\right) \right\|_2^2 - \sum_{j=1}^{K} \left\| (I - C_j \circ \mathcal{L}_h^a)\left(u_h - u_h^{(t)}\right) \right\|_2^2 \mathbf{1}_{r(a_h, f_h, u_h^{(t)}) \neq j}
$$

$$
+ \sum_{j=1}^{K} \mathbf{1}_{r(a_h, f_h, u_h^{(t)}) \neq j} \sum_{k=1}^{K} \left\| (I - C_k \circ \mathcal{L}_h^a)\left(u_h - u_h^{(t)}\right) \right\|_2^2
$$

$$
- (K-1) \sum_{k=1}^{K} \left\| (I - C_k \circ \mathcal{L}_h^a)\left(u_h - u_h^{(t)}\right) \right\|_2^2
$$

$$
= \sum_{j=1}^{K} \left( \sum_{k=1}^{K} \left\| (I - C_k \circ \mathcal{L}_h^a)\left(u_h - u_h^{(t)}\right) \right\|_2^2 - \left\| (I - C_j \circ \mathcal{L}_h^a)\left(u_h - u_h^{(t)}\right) \right\|_2^2 \right) \mathbf{1}_{r(a_h, f_h, u_h^{(t)}) \neq j}
$$

$$
- (K-2) \sum_{k=1}^{K} \left\| (I - C_k \circ \mathcal{L}_h^a)\left(u_h - u_h^{(t)}\right) \right\|_2^2
$$

$$
= \sum_{j=1}^{K} \sum_{k=1}^{K} \left\| (I - C_k \circ \mathcal{L}_h^a)\left(u_h - u_h^{(t)}\right) \right\|_2^2 \mathbf{1}_{k \neq j} \mathbf{1}_{r(a_h, f_h, u_h^{(t)}) \neq j}
$$

$$
- (K-2) \sum_{k=1}^{K} \left\| (I - C_k \circ \mathcal{L}_h^a)\left(u_h - u_h^{(t)}\right) \right\|_2^2
$$

$\square$

## C.2 PROOF OF PROPOSITION C.2

**Proposition C.2.** *For any router $r$ defined by $r(a, f, u^{(t)}) = argmax_{j \in [K]} g_j(a, f, u^{(t)})$, any $a_h \in \mathcal{A}$, $f_h \in \mathcal{F}$, and $u_h^{(t)}, u_h \in \mathcal{U}$, the routing loss $l_{route}$ satisfies:*

$$\log(2) l_{route}\left(r, a_h, f_h, u_h^{(t)}, u_h\right) \leq \Psi(\mathbf{g}, a_h, f_h, u_h^{(t)}, u_h)$$

*Proof.* By Lemma C.1, we know that

$$
\log(2) l_{route}\left(r, a_h, f_h, u_h^{(t)}, u_h\right) = \log(2) \sum_{j=1}^{K} \sum_{k=1}^{K} \left\| (I - C_k \circ \mathcal{L}_h^a)\left(u_h - u_h^{(t)}\right) \right\|_2^2 \mathbf{1}_{k \neq j} \mathbf{1}_{r(a_h, f_h, u_h^{(t)}) \neq j}
$$

$$
- \log(2)(K-2) \sum_{j=1}^{K} \left\| (I - C_j \circ \mathcal{L}_h^a)\left(u_h - u_h^{(t)}\right) \right\|_2^2
$$

$$
\leq \log(2) \sum_{j=1}^{K} \sum_{k=1}^{K} \left\| (I - C_k \circ \mathcal{L}_h^a)\left(u_h - u_h^{(t)}\right) \right\|_2^2 \mathbf{1}_{k \neq j} \mathbf{1}_{r(a_h, f_h, u_h^{(t)}) \neq j}
$$

$$
\overset{(a)}{\leq} - \sum_{j=1}^{K} \sum_{k=1}^{K} \left\| (I - C_k \circ \mathcal{L}_h^a)\left(u_h - u_h^{(t)}\right) \right\|_2^2 \mathbf{1}_{k \neq j} \log\left( \frac{\exp\left(g_j(a, f, u^{(t)})\right)}{\sum_{k=1}^{K} \exp\left(g_k(a, f, u^{(t)})\right)} \right)
$$

$$
= \Psi(\mathbf{g}, a_h, f_h, u_h^{(t)}, u_h)
$$

(a) if $r(a_h, f_h, u_h^{(t)}) \neq j$, $\frac{\exp\left(g_j(a, f, u^{(t)})\right)}{\sum_{k=1}^{K} \exp\left(g_k(a, f, u^{(t)})\right)} < 0.5$ which implies that $- \log\left( \frac{\exp\left(g_j(a, f, u^{(t)})\right)}{\sum_{k=1}^{K} \exp\left(g_k(a, f, u^{(t)})\right)} \right) \geq \log(2) \mathbf{1}_{r(a_h, f_h, u_h^{(t)}) \neq j}$

$\square$

## C.3 PROOF OF THEOREM 5.1

**Theorem 5.1.** *Let $\tilde{c}_j(a_h, u_h^{(t)}, u_h) < \bar{E} < \infty$ for all $j \in [K]$. If there exists $j \in [K]$ such that $\tilde{c}_j(a_h, u_h^{(t)}, u_h) > E_{min} > 0$, then, for any collection of solvers $\{C_j\}_{j=1}^{K}$ and linear discrete operator $\mathcal{L}_h^a$, $\Psi$ is Bayes consistent surrogate for $l_{route}$.*

*Proof.* For a given $a_h, f_h$, let $u_h$ be $\mathcal{G}_h(a_h, f_h)$ where $\mathcal{G}_h$ denotes the solution operator acting on the grid $G_h$. Furthermore, let's consider routers of the form

$$r(a, f, u^{(t)}) = argmax_{j \in [K]} g_j(a, f, u^{(t)})$$

For a given $a_h, f_h, u_h^{(t)} \in \mathcal{A} \times \mathcal{F} \times \mathcal{U}$, let the optimal loss under $l_{route}$ be $l_{route}^*\left(a_h, f_h, u_h^{(t)}\right) = \inf_{\tilde{r}} l_{route}\left(\tilde{r}, a_h, f_h, u_h^{(t)}, \mathcal{G}_h(a_h, f_h)\right)$. Similarly, let the optimal loss under $\Psi$ be $\Psi^*\left(a_h, f_h, u_h^{(t)}\right) = \inf_{\tilde{\mathbf{g}}} \Psi\left(\tilde{\mathbf{g}}, a_h, f_h, u_h^{(t)}, \mathcal{G}_h(a_h, f_h)\right)$. Let $B_j(a_h, f_h, u_h^{(t)}) = \sum_{k=1}^{K} \left\| (I - C_k \circ \mathcal{L}_h^a)\left(\mathcal{G}_h(a_h, f_h) - u_h^{(t)}\right) \right\|_2^2 \mathbf{1}_{k \neq j}$.

$$l_{\text{route}}\left(r, a_h, f_h, u_h^{(t)}, \mathcal{G}_h\left(a_h, f_h\right)\right) - l_{\text{route}}^*\left(a_h, f_h, u_h^{(t)}\right)$$

$$\overset{(a)}{=} \sum_{j=1}^K \sum_{k=1}^K \left\| (I - C_k \circ \mathcal{L}_h^a)\left(u_h - u_h^{(t)}\right) \right\|_2^2 \mathbf{1}_{k \neq j} \mathbf{1}_{r(a_h, f_h, u_h^{(t)}) \neq j} - (K-2) \sum_{j=1}^K \left\| (I - C_j \circ \mathcal{L}_h^a)\left(u_h - u_h^{(t)}\right) \right\|_2^2$$

$$- \inf_{\tilde{r}} \sum_{j=1}^K \sum_{k=1}^K \left\| (I - C_k \circ \mathcal{L}_h^a)\left(u_h - u_h^{(t)}\right) \right\|_2^2 \mathbf{1}_{k \neq j} \mathbf{1}_{\tilde{r}(a_h, f_h, u_h^{(t)}) \neq j} + (K-2) \sum_{j=1}^K \left\| (I - C_j \circ \mathcal{L}_h^a)\left(u_h - u_h^{(t)}\right) \right\|_2^2$$

$$= \sum_{j=1}^K B_j(a_h, f_h, u_h^{(t)}) \mathbf{1}_{r(a_h, f_h, u_h^{(t)}) \neq j} - \inf_{\tilde{r}} \sum_{j=1}^K B_j(a_h, f_h, u_h^{(t)}) \mathbf{1}_{\tilde{r}(a_h, f_h, u_h^{(t)}) \neq j}$$

$$= \sum_{k=1}^K B_k(a_h, f_h, u_h^{(t)}) \left( \sum_{j=1}^K \frac{B_j(a_h, f_h, u_h^{(t)})}{\sum_{k=1}^K B_k(a_h, f_h, u_h^{(t)})} \mathbf{1}_{r(a_h, f_h, u_h^{(t)}) \neq j} - \inf_{\tilde{r}} \sum_{j=1}^K \frac{B_j(a_h, f_h, u_h^{(t)})}{\sum_{k=1}^K B_k(a_h, f_h, u_h^{(t)})} \mathbf{1}_{\tilde{r}(a_h, f_h, u_h^{(t)}) \neq j} \right)$$

(a) by Lemma C.1

Let $\mathcal{X} = \mathcal{A} \times \mathcal{F} \times \mathcal{U}$ and $\mathcal{Y} = [K]$. Let $\mathcal{P}_\mathcal{X}$ denote the degenerate distribution suported at the point $(a_h, f_h, u_h^{(t)})$. We define the conditional distribtion - $P(Y = j \mid X = (a_h, f_h, u_h^{(t)})) = \frac{B_j(a_h, f_h, u_h^{(t)})}{\sum_{k=1}^K B_k(a_h, f_h, u_h^{(t)})}$ for $j \in [K]$. The risk and optimal risk of $0 - 1$ loss under this distribution can be written as:

$$\mathcal{R}_{0-1}(r) = \sum_{j=1}^K \frac{B_j(a_h, f_h, u_h^{(t)})}{\sum_{k=1}^K B_k(a_h, f_h, u_h^{(t)})} \mathbf{1}_{r(a_h, f_h, u_h^{(t)}) \neq j}$$

$$\mathcal{R}_{0-1}^* = \inf_r \sum_{j=1}^K \frac{B_j(a_h, f_h, u_h^{(t)})}{\sum_{k=1}^K B_k(a_h, f_h, u_h^{(t)})} \mathbf{1}_{r(a_h, f_h, u_h^{(t)}) \neq j}$$

If $r(a_h, f_h, u_h^{(t)}) = \arg\max_{j \in [k]} g_j(a_h, f_h, u_h^{(t)})$ for all $x \in \mathcal{X}$, then the he risk and optimal risk of cross entropy loss ($l_{ce}(\mathbf{g}, x, y) - \log\left(\frac{\exp(g_y(x))}{\sum_{k=1}^K \exp(g_k(x))}\right)$) under this distribution can be written as:

$$\mathcal{R}_{ce}(\mathbf{g}) = -\sum_{j=1}^K \frac{B_j(a_h, f_h, u_h^{(t)})}{\sum_{k=1}^K B_k(a_h, f_h, u_h^{(t)})} \log\left( \frac{\exp\left(g_j(a_h, f_h, u_h^{(t)})\right)}{\sum_{k=1}^K \exp\left(g_k(a_h, f_h, u_h^{(t)})\right)} \right)$$

$$\mathcal{R}_{ce}^* = \inf_{\mathbf{g}} -\sum_{j=1}^K \frac{B_j(a_h, f_h, u_h^{(t)})}{\sum_{k=1}^K B_k(a_h, f_h, u_h^{(t)})} \log\left( \frac{\exp\left(g_j(a_h, f_h, u_h^{(t)})\right)}{\sum_{k=1}^K \exp\left(g_k(a_h, f_h, u_h^{(t)})\right)} \right)$$

From Theorem 3.1 of (Mao et al., 2023), $\mathcal{R}_{0-1}(r) - \mathcal{R}_{0-1}^* \leq \Gamma^{-1}\left(\mathcal{R}_{ce}(\mathbf{g}) - \mathcal{R}_{ce}^*\right)$ if $r(a_h, f_h, u_h^{(t)}) = \arg\max_{j \in [k]} g_j(a_h, f_h, u_h^{(t)})$ where $\Gamma(z) = \frac{1+z}{2}\log(1+z) + \frac{1-z}{2}\log(1-z)$. Then,

$$l_{\text{route}}\left(r, a_h, f_h, u_h^{(t)}, \mathcal{G}_h\left(a_h, f_h\right)\right) - l_{\text{route}}^*\left(a_h, f_h, u_h^{(t)}\right)$$

$$= \sum_{k=1}^{K} B_k(a_h, f_h, u_h^{(t)}) \left( \sum_{j=1}^{K} \frac{B_j(a_h, f_h, u_h^{(t)})}{\sum_{k=1}^{K} B_k(a_h, f_h, u_h^{(t)})} \mathbf{1}_{r(a_h, f_h, u_h^{(t)}) \neq j} - \inf_{\tilde{r}} \sum_{j=1}^{K} \frac{B_j(a_h, f_h, u_h^{(t)})}{\sum_{k=1}^{K} B_k(a_h, f_h, u_h^{(t)})} \mathbf{1}_{\tilde{r}(a_h, f_h, u_h^{(t)}) \neq j} \right)$$

$$\leq \sum_{k=1}^{K} B_k(a_h, f_h, u_h^{(t)}) \Gamma^{-1} \left( -\sum_{j=1}^{K} \frac{B_j(a_h, f_h, u_h^{(t)})}{\sum_{k=1}^{K} B_k(a_h, f_h, u_h^{(t)})} \log \left( \frac{\exp\left(g_j(a_h, f_h, u_h^{(t)})\right)}{\sum_{k=1}^{K} \exp\left(g_k(a_h, f_h, u_h^{(t)})\right)} \right) \right.$$

$$\left. - \inf_{\mathbf{g}} - \sum_{j=1}^{K} \frac{B_j(a_h, f_h, u_h^{(t)})}{\sum_{k=1}^{K} B_k(a_h, f_h, u_h^{(t)})} \log \left( \frac{\exp\left(g_j(a_h, f_h, u_h^{(t)})\right)}{\sum_{k=1}^{K} \exp\left(g_k(a_h, f_h, u_h^{(t)})\right)} \right) \right)$$

$$\overset{(a)}{\leq} \bar{E} K (K-1) \Gamma^{-1} \left( -\sum_{j=1}^{K} \frac{B_j(a_h, f_h, u_h^{(t)})}{\sum_{k=1}^{K} B_k(a_h, f_h, u_h^{(t)})} \log \left( \frac{\exp\left(g_j(a_h, f_h, u_h^{(t)})\right)}{\sum_{k=1}^{K} \exp\left(g_k(a_h, f_h, u_h^{(t)})\right)} \right) \right.$$

$$\left. - \inf_{\mathbf{g}} - \sum_{j=1}^{K} \frac{B_j(a_h, f_h, u_h^{(t)})}{\sum_{k=1}^{K} B_k(a_h, f_h, u_h^{(t)})} \log \left( \frac{\exp\left(g_j(a_h, f_h, u_h^{(t)})\right)}{\sum_{k=1}^{K} \exp\left(g_k(a_h, f_h, u_h^{(t)})\right)} \right) \right)$$

$$\overset{(b)}{\leq} \bar{E} K (K-1) \Gamma^{-1} \left( -\sum_{j=1}^{K} \frac{B_j(a_h, f_h, u_h^{(t)})}{(K-1)E_{min}} \log \left( \frac{\exp\left(g_j(a_h, f_h, u_h^{(t)})\right)}{\sum_{k=1}^{K} \exp\left(g_k(a_h, f_h, u_h^{(t)})\right)} \right) \right.$$

$$\left. - \inf_{\mathbf{g}} - \sum_{j=1}^{K} \frac{B_j(a_h, f_h, u_h^{(t)})}{(K-1)E_{min}} \log \left( \frac{\exp\left(g_j(a_h, f_h, u_h^{(t)})\right)}{\sum_{k=1}^{K} \exp\left(g_k(a_h, f_h, u_h^{(t)})\right)} \right) \right)$$

$$= \bar{E} K (K-1) \Gamma^{-1} \left( \frac{\Psi\left(\mathbf{g}, a_h, f_h, u_h^{(t)}, \mathcal{G}_h\left(a_h, f_h\right)\right) - \Psi^*\left(a_h, f_h, u_h^{(t)}\right)}{(K-1)E_{min}} \right)$$

(a) since $\left\| \left(I - C_j \circ \mathcal{L}_h^a\right)\left(e_h^{(t)}\right) \right\|_2^2 < \bar{E}$ for all $j \in [K]$, (b) since $\Gamma^{-1}$ is non-decreasing and $\exists j \in [K]$ such that $\left\| \left(I - C_j \circ \mathcal{L}_h^a\right)\left(e_h^{(t)}\right) \right\|_2^2 > E_{min}$

Finally,

$$\lim_{n \to \infty} \mathcal{R}_{\text{route}}(r_n) - \mathcal{R}_{\text{route}}^*$$

$$\stackrel{(a)}{=} \lim_{n \to \infty} \mathbb{E}_{a_h, f_h \sim \mathcal{P}_{\mathcal{A} \times \mathcal{F}}} \left[ l_{\text{route}}\left(r_n, a_h, f_h, u_h^{(t)}, \mathcal{G}_h(a_h, f_h)\right) - l_{\text{route}}^*\left(a_h, f_h, u_h^{(t)}\right) \right]$$

$$\leq \lim_{n \to \infty} \mathbb{E}_{a_h, f_h \sim \mathcal{P}_{\mathcal{A} \times \mathcal{F}}} \left[ \bar{E} K (K-1) \Gamma^{-1} \left( \frac{\Psi\left(\tilde{\mathbf{g}}, a_h, f_h, u_h^{(t)}, \mathcal{G}_h(a_h, f_h)\right) - \Psi^*\left(a_h, f_h, u_h^{(t)}\right)}{(K-1) E_{min}} \right) \right]$$

$$\stackrel{(b)}{\leq} \lim_{n \to \infty} \bar{E} K (K-1) \Gamma^{-1} \left( \frac{\mathbb{E}_{a_h, f_h \sim \mathcal{P}_{\mathcal{A} \times \mathcal{F}}} \left[ \Psi\left(\mathbf{g}_n, a_h, f_h, u_h^{(t)}, \mathcal{G}_h(a_h, f_h)\right) - \Psi^*\left(a_h, f_h, u_h^{(t)}\right) \right]}{(K-1) E_{min}} \right)$$

$$= \lim_{n \to \infty} \bar{E} K (K-1) \Gamma^{-1} \left( \frac{\mathcal{R}_\Psi(\mathbf{g}_n) - \mathcal{R}_\Psi^*}{(K-1) E_{min}} \right)$$

$$\stackrel{(c)}{=} \bar{E} K (K-1) \Gamma^{-1} \left( \frac{\lim_{n \to \infty} \mathcal{R}_\Psi(\mathbf{g}_n) - \mathcal{R}_\Psi^*}{(K-1) E_{min}} \right)$$

$$= \bar{E} K (K-1) \Gamma^{-1} (0)$$

$$\stackrel{(d)}{=} 0$$

(a) $\mathcal{R}_{\text{route}}^* = \mathbb{E}_{a_h, f_h \sim \mathcal{P}_{\mathcal{A} \times \mathcal{F}}} \left[ l_{\text{route}}^*\left(a_h, f_h, u_h^{(t)}\right) \right]$ since the infimum is taken over all measurable functions, (b) by Jensen's inequality since $\Gamma^{-1}$ is concave, (c) by continuity of $\Gamma^{-1}$ at 0, (d) $\Gamma^{-1}(0) = 0$

$$\square$$

## D  TRAINING DETAILS

Data for both DeepONet and the routers is sampled from a zero-mean Gaussian Random Field on the periodic domain with covariance operator $(-\Delta + 9I)^{-2}$ as mentioned in Section 7. We do this by generating samples in Fourier space: for each non-zero mode $k$, we draw an independent complex coefficient from a Gaussian distribution with mean 0 and variance $(4\pi^2 \|k\|_2^2 + 9)^{-2}$, enforce a Hermitian symmetry to obtain a real-valued field, set the DC mode to 0 to ensure zero mean for Poisson, and apply inverse Discrete Fourier Transform to obtain the field in physical space. For each sample, we compute reference solutions with a least squares solver and treat them as ground truth.

This data is used to trained our DeepONet models and LSTM routers. All the models were implemented using PyTorch and all the models were trained on an Nvidia A40 GPU.

Table 3 contains all hyperparameter details for the DeepONet. DeepONet took 30 minutes to train. We then use the model with the best validation loss.

The routers are LSTM models trained with scheduled sampling. We use a warm-up of $e_w$ epochs with teacher-forcing probability $p_{tf}(e) = ss_{\text{start}}$. After the warm-up, the $p_{tf}$ decays geomterically by a factor of $\gamma_{tf} < 1$ per epoch and is floored by $s_{\text{end}}$:

$$p_{tf}(e) = \begin{cases} ss_{\text{start}} & e \leq e_w \\ \max(ss_{start} \gamma_{tf}^{e - e_w}, ss_{end}) & e > e_w \end{cases}$$

At each time step, with probability $p_{tf}(e)$, we feed the teacher-forced greedy iterate; otherwise, we feed the router's own predicted iterate.

Since LSTMs on long rollouts can suffer from exploding/vanishing gradients, we use truncated backpropagation through time (TBPTT) (Mozer, 2013; Robinson & Fallside, 1987; Werbos, 1988): the forward pass unrolls the entire trajectory, but gradients are propagated only through the most recent $w_{\text{bptt}}(e)$ steps at epoch $e$. Hidden states are passed forward between segments, while earlier segments are treated as stop-gradient.

| Hyperparameter | Value |
|---|---|
| Learning rate | 1e-3 |
| Branch Dimension | 64 |
| Hidden dimension for branch net | 128 |
| No. of hidden layers in branch net | 2 |
| Hidden dimension for trunk net | 128 |
| No. of hidden layers in trunk net | 2 |
| Gradient Clipping Norm | 1.0 |
| Weight Decay | 0.005 |
| Batch size | 128 |
| Training samples | 15000 |
| Validation samples | 3000 |
| Epochs | 100 |

Table 3: Hyperparameter settings for DeepONet

| Hyperparameter | Value |
|---|---|
| Learning rate | 1e-3 |
| Branch Dimension | 64 |
| Hidden dimension | 64 |
| No. of hidden layers | 3 |
| Gradient Clipping Norm | 1.0 |
| Weight Decay | 0.005 |
| Batch size | 32 |
| Training samples | 64 |
| Validation samples | 32 |
| Epochs | 100 |
| $ss_{\text{start}}$ | 1.0 |
| $\gamma_{tf}$ | 0.95 |
| $ss_{\text{end}}$ | 0.0 |
| $w_{\text{start}}$ | $0.1T_{\max}$ |
| $\gamma_{\text{bptt}}$ | 1.25 |
| $e_w$ | 10 |
| $f_{\text{bptt}}$ | 4 |

Table 4: Hyperparameter settings for routers

We employ a curriculum learning approach analogous to scheduled sampling. Let $T_{\max}$ be the horizon (300 for Jacobi/GS and 100 for MG). With a warm-up of $e_w$ epochs,

$$w_{\text{bptt}}(e) = \begin{cases} w_{\text{start}} & e \leq e_w \\ \min\left(T_{\max}, w_{\text{start}}\gamma_{\text{bptt}}^{\left\lfloor \frac{e-e_w}{f_{\text{bptt}}} \right\rfloor}\right) & e > e_w \end{cases} \tag{12}$$

so the window grows geometrically by a factor of $\gamma_{\text{bptt}} > 1$ every $f_{\text{bptt}}$ epochs and is capped at the full trajectory length.

Table 4 contains all hyperparameter details for the LSTM routers. The routers took a maximum of 4 hours and 30 minutes to train. We then use the model with the best validation loss for testing. Data-related details in Table 4 apply to all of our trained routers except the routers for the experiment with increasing $K$ which were trained with 1024 training samples and 128 validations samples to encourage the model to learn some of the nuanced differences between the classes.

## E  ADDITIONAL EXPERIMENTAL RESULTS

### E.1  CONVERGENCE HISTORIES

See Figure 2

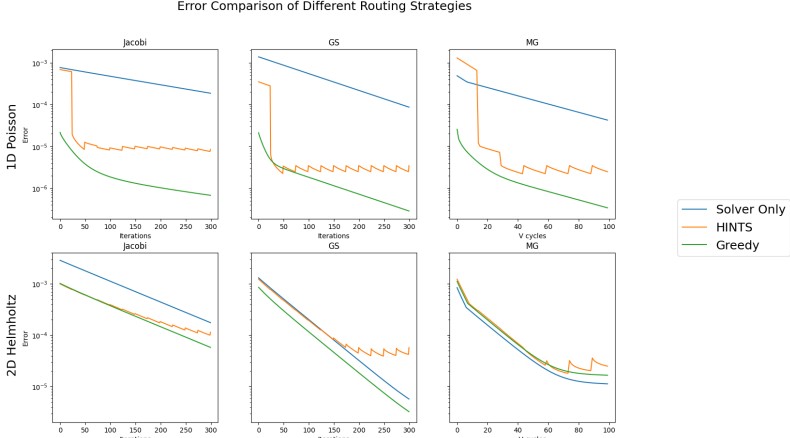

Figure 2: Convergence histories for representative test instances. Rows: 1D Poisson (top) and 2D Helmholtz (bottom). Columns: Jacobi, Gauss–Seidel (GS), and multigrid (MG). Greedy yields near-monotone decay and the lowest errors, whereas HINTS shows sawtooth behaviors.

| Equation | 1D Poisson | | 2D Poisson | | 1D Helmholtz | | 2D Helmholtz | |
|---|---|---|---|---|---|---|---|---|
| Methods | $\|\mathcal{L}_h^a e_h^{(T)}\|$ | AUC | $\|\mathcal{L}_h^a e_h^{(T)}\|$ | AUC | $\|\mathcal{L}_h^a e_h^{(T)}\|$ | AUC | $\|\mathcal{L}_h^a e_h^{(T)}\|$ | AUC |
| Jacobi-related Solvers | | | | | | | | |
| Jacobi Only | 7.775 (4.237) | 5156.608 (2729.501) | 2.842 (1.124) | 4979.462 (1570.207) | 8.89 (4.62) | 5977.502 (3061.518) | 2.653 (1.049) | 4885.359 (1536.61) |
| HINTS-Jacobi | 4.871 (1.526) | 2511.313 (1113.55) | 94.061 (0.431) | 6194.836 (480.136) | 23.035 (12.824) | 6860.808 (3824.142) | 24.066 (0.258) | 5486.311 (1491.338) |
| Greedy-Jacobi | 5.684 (4.186) | 2505.155 (1638.618) | 1.225 (0.643) | 2097.546 (468.004) | 14.339 (13.896) | 5967.264 (5235.896) | 2.653 (1.049) | 4885.359 (1536.61) |
| GS-related Solvers | | | | | | | | |
| GS Only | 2.001 (1.091) | 3272.362 (1741.517) | 0.202 (0.08) | 2686.598 (858.228) | 2.209 (1.15) | 3767.204 (1933.993) | 0.176 (0.07) | 2625.785 (836.048) |
| HINTS-GS | 2.749 (0.001) | 904.028 (394.014) | 115.382 (0.001) | 4945.064 (339.648) | 5.727 (3.043) | 1193.716 (487.82) | 23.431 (0.02) | 3181.072 (817.491) |
| Greedy-GS | 0.012 (0.007) | 170.604 (57.765) | 0.027 (0.008) | 998.784 (193.159) | 0.035 (0.015) | 250.746 (89.774) | 0.176 (0.07) | 2625.785 (836.048) |
| MG-related Solvers | | | | | | | | |
| MG Only | 1.961 (0.541) | 819.828 (292.076) | 0.093 (0.022) | 460.981 (111.146) | 2.017 (0.529) | 899.308 (313.69) | 0.081 (0.02) | 448.77 (108.006) |
| HINTS-MG | 0.138 | 354.601 (149.921) | 3.246 | 1414.353 (73.994) | 0.49 (0.196) | 451.287 (178.914) | 0.653 (0.052) | 640.516 (101.671) |
| Greedy-MG | 0.019 (0.012) | 61.195 (17.887) | 0.022 (0.005) | 284.36 (51.462) | 0.053 (0.022) | 99.709 (37.971) | 0.081 (0.02) | 448.77 (108.006) |

Table 5: Final residual and AUC of squared $L^2$ residual (lower is better). Values are mean ($\pm$ standard error (s.e.)) over 64 test instances; both mean and s.e. are reported in $\times 10^{-3}$. If a standard error is not shown, it is $< 10^{-3}$ in the reported units (raw $< 10^{-6}$). Bold indicates the best method within each solver family.

## E.2 RESIDUAL COMPARISON

Table 5 summarizes the performance of single-solver schedules, HINTS, and greedy with respect to the final residuals $r_h^{(T)} = \|f_h - \mathcal{L}_h^a u_h^{(T)}\|$ or $\|\mathcal{L}_h^a e_h^{(T)}\|$ and its AUC $AUC_T = \sum_{t=1}^{T} \|r_h^{(t)}\|_2^2$. Greedy outperforms its HINTS and single-solver counterparts in most equations. We must note that our greedy router is trained to reduce error, not residual. The same error can induce very different residuals depending on the spectrum $\mathcal{L}_h^a$. Table 6 exhibits how residuals are affected by the number of solvers in the solver ensemble. Similar to error, we observe both the final residual and AUC decrease as the number of solvers increase.

| Equation | 1D Poisson | | 1D Helmholtz | |
|---|---|---|---|---|
| # of Solvers | $\|\mathcal{L}_h^a e_h^{(T)}\|$ | AUC | $\|\mathcal{L}_h^a e_h^{(T)}\|$ | AUC |
| 2 | 0.121 (0.055) | 473.906 (158.205) | 0.321 (0.113) | 679.886 (252.811) |
| 3 | 0.078 (0.042) | 360.389 (119.802) | 0.21 (0.08) | 530.239 (195.882) |
| 4 | 0.067 (0.038) | 328.62 (109.061) | 0.181 (0.072) | 490.084 (183.068) |
| 5 | 0.061 (0.035) | 319.509 (105.729) | 0.166 (0.067) | 470.193 (176.504) |
| 6 | 0.045 (0.027) | 288.867 (96.794) | 0.165 (0.067) | 487.057 (192.411) |

Table 6: Final residual and AUC of squared $L^2$ residual for varying numbers of solvers. Values are mean ($\pm$ standard error (s.e.)) over 64 test instances; both mean and s.e. are reported in $\times 10^{-3}$.

| Equation | 1D Poisson | | | | | | 2D Poisson | | | | | |
|---|---|---|---|---|---|---|---|---|---|---|---|---|
| Methods/ Mode | Mode 1 Error | Mode 1 AUC | Mode 5 Error | Mode 5 AUC | Mode 10 Error | Mode 10 AUC | Mode 1 Error | Mode 1 AUC | Mode 5 Error | Mode 5 AUC | Mode 10 Error | Mode 10 AUC |
| Jacobi-related Solvers | | | | | | | | | | | | |
| Jacobi Only | 1.124 (0.612) | 732.713 (399.296) | - | **0.076 (0.043)** | - | **0.001** | 0.03 (0.014) | 382.094 (183.783) | - | **0.012 (0.006)** | - | - |
| HINTS-Jacobi | 0.059 (0.024) | 128.076 (64.871) | 0.001 (0.001) | 0.348 (0.121) | - | 0.041 (0.021) | 0.296 (0.003) | 227.594 (78.715) | 0.003 | 0.169 (0.055) | 0.001 | 0.03 (0.009) |
| Greedy-Jacobi | **0.006 (0.004)** | **4.152 (2.593)** | - | 0.192 (0.094) | - | 0.025 (0.016) | **0.009 (0.004)** | **114.046 (54.747)** | - | 0.105 (0.057) | - | 0.014 (0.008) |
| GS-related Solvers | | | | | | | | | | | | |
| GS Only | 0.285 (0.155) | 458.834 (250.064) | - | 0.223 (0.136) | - | 0.096 (0.072) | - | 196.347 (94.367) | - | 0.014 (0.007) | - | **0.001** |
| HINTS-GS | 0.017 | 100.582 (52.598) | 0.001 | 0.176 (0.049) | - | 0.049 (0.028) | 0.258 | 168.877 (59.252) | 0.003 | 0.162 (0.032) | 0.001 | 0.052 (0.006) |
| Greedy-GS | **0.002 (0.001)** | **2.661 (1.595)** | - | **0.12 (0.058)** | - | **0.026 (0.015)** | - | **62.858 (30.237)** | - | **0.09 (0.044)** | - | 0.014 (0.007) |
| Multigrid methods | | | | | | | | | | | | |
| MG Only | 0.282 (0.078) | 116.885 (42.581) | - | 0.043 (0.022) | - | 0.013 (0.007) | 0.001 | 25.497 (11.766) | - | **0.003 (0.001)** | - | - |
| HINTS-MG | 0.014 | 43.405 (20.908) | - | **0.027 (0.006)** | - | 0.009 (0.003) | 0.078 | 31.289 (10.201) | - | 0.035 (0.004) | - | 0.01 (0.001) |
| Greedy-MG | **0.003 (0.002)** | **0.999 (0.622)** | - | 0.029 (0.013) | - | **0.008 (0.005)** | - | **13.358 (6.045)** | - | 0.031 (0.016) | - | 0.007 (0.004) |

Table 7: Final error and AUC of squared $L^2$ error for Mode 1, 5, and 10 (lower is better) for 1D/2D Poisson. Values are mean ($\pm$ standard error (s.e.)) over 64 test instances; both mean and s.e. are reported in $\times 10^{-3}$. If a standard error is not shown, it is $< 10^{-3}$ in the reported units (raw $< 10^{-6}$). Bold indicates the best method within each solver family.

| Equation | 1D Helmholtz | | | | | | 2D Helmholtz | | | | | |
|---|---|---|---|---|---|---|---|---|---|---|---|---|
| Methods/ Mode | Mode 1 Error | Mode 1 AUC | Mode 5 Error | Mode 5 AUC | Mode 10 Error | Mode 10 AUC | Mode 1 Error | Mode 1 AUC | Mode 5 Error | Mode 5 AUC | Mode 10 Error | Mode 10 AUC |
| Jacobi-related Solvers | | | | | | | | | | | | |
| Jacobi Only | 1.253 (0.651) | 835.102 (434.018) | - | **0.078 (0.044)** | - | **0.001** | 0.028 (0.013) | 372.63 (179.23) | - | **0.012 (0.006)** | - | - |
| HINTS-Jacobi | 0.099 (0.047) | 140.399 (63.534) | 0.004 (0.002) | 0.576 (0.2) | 0.001 (0.001) | 0.088 (0.044) | 0.493 (0.025) | 427.437 (170.326) | 0.007 | 0.19 (0.006) | 0.002 | 0.026 |
| Greedy-Jacobi | **0.023 (0.027)** | **15.089 (13.203)** | - | 0.577 (0.635) | - | 0.078 (0.057) | **0.028 (0.013)** | **372.63 (179.23)** | - | **0.012 (0.006)** | - | - |
| GS-related Solvers | | | | | | | | | | | | |
| GS Only | 0.307 (0.159) | 519.339 (269.921) | - | 0.242 (0.146) | - | 0.105 (0.078) | - | **191.408 (91.996)** | - | **0.014 (0.007)** | - | 0.001 |
| HINTS-GS | 0.07 (0.028) | 127.525 (59.386) | 0.003 (0.002) | 0.296 (0.09) | 0.001 (0.001) | 0.105 (0.052) | 0.309 | 233.092 (88.614) | 0.007 | 0.191 (0.007) | 0.002 | 0.024 |
| Greedy-GS | **0.004 (0.002)** | **7.034 (3.179)** | - | **0.179 (0.1)** | - | **0.058 (0.036)** | - | **191.408 (91.996)** | - | **0.014 (0.007)** | - | 0.001 |
| Multigrid methods | | | | | | | | | | | | |
| MG Only | 0.283 (0.074) | 125.54 (44.397) | - | 0.043 (0.022) | - | **0.013 (0.007)** | 0.001 | 24.863 (11.508) | - | **0.002 (0.001)** | - | - |
| HINTS-MG | 0.055 (0.022) | 52.357 (23.002) | - | 0.054 (0.02) | - | 0.023 (0.009) | 0.074 (0.001) | 32.657 (11.313) | - | 0.051 (0.001) | - | 0.009 |
| Greedy-MG | **0.007 (0.003)** | **2.678 (1.159)** | - | **0.041 (0.019)** | - | 0.016 (0.011) | **0.001** | **24.863 (11.508)** | - | **0.002 (0.001)** | - | - |

Table 8: Final error and AUC of squared $L^2$ error for Mode 1, 5, and 10 (lower is better) for 1D/2D Helmholtz. Values are mean ($\pm$ standard error (s.e.)) over 64 test instances; both mean and s.e. are reported in $\times 10^{-3}$. If a standard error is not shown, it is $< 10^{-3}$ in the reported units (raw $< 10^{-6}$). Bold indicates the best method within each solver family.

| Equation | 1D Poisson | | | | | | 1D Helmholtz | | | | | |
|---|---|---|---|---|---|---|---|---|---|---|---|---|
| # of Solvers/ Mode | Mode 1 Error | Mode 1 AUC | Mode 5 Error | Mode 5 AUC | Mode 10 Error | Mode 10 AUC | Mode 1 Error | Mode 1 AUC | Mode 5 Error | Mode 5 AUC | Mode 10 Error | Mode 10 AUC |
| 2 | 0.013 (0.008) | 5.549 (3.466) | - | 0.396 (0.194) | - | 0.057 (0.029) | 0.035 (0.015) | 15.082 (6.582) | - | 0.608 (0.352) | - | 0.115 (0.082) |
| 3 | 0.01 (0.006) | 5.007 (3.126) | - | 0.293 (0.143) | - | 0.04 (0.025) | 0.027 (0.012) | 13.577 (5.925) | - | 0.449 (0.26) | - | 0.082 (0.058) |
| 4 | 0.009 (0.006) | 4.779 (2.984) | - | 0.263 (0.128) | - | 0.036 (0.023) | 0.024 (0.011) | 12.943 (5.648) | - | 0.399 (0.231) | - | 0.072 (0.051) |
| 5 | 0.008 (0.005) | 4.654 (2.906) | 0.0 (0.0) | 0.257 (0.125) | 0.0 (0.0) | 0.036 (0.023) | 0.022 (0.01) | 12.571 (5.486) | - | 0.376 (0.217) | - | 0.068 (0.049) |
| 6 | 0.006 (0.004) | 4.174 (2.606) | - | 0.212 (0.104) | - | 0.029 (0.018) | 0.022 (0.01) | 12.535 (5.47) | - | 0.355 (0.205) | - | 0.061 (0.043) |

Table 9: Final error and AUC of squared $L^2$ error of Mode 1, 5, and 10 for varying numbers of solvers. Values are mean ($\pm$ standard error (s.e.)) over 64 test instances; both mean and s.e. are reported in $\times 10^{-3}$.

### E.3 FOURIER MODE-WISE ERROR COMPARISON

We assess frequency-resolved performance by projecting the error onto the discrete Fourier basis. Tables 7 and 8 report, for modes 1, 5, and 10, the mode-wise final error and mode-wise AUC, comparing single-solver baselines, HINTS, and the greedy router. As a result of including a deep learning model, Greedy consistently achieves the smallest mode-1 error/AUC across equations and solver families. For modes 5 and 10, single-solver schedules sometimes have an edge, reflecting the tendency of classical smoothers to damp high-frequency components more aggressively than ML surrogates (spectral bias). Overall, greedy delivers more uniform convergence across the spectrum: it routes to whichever solver most decreases the full $L^2$ error, and by Parseval's identity $|e_h^{(t)}|_2^2 = \sum_m |\hat{u}_m^{(t)} - \hat{u}_m|^2$, reductions in the objective correspond to reducing energy across all modes rather than giving preferential treatment to a subset. Additionally, in Table 9, we observe that all mode-wise errors/AUCs reduce with the inclusion of more solvers.

## F    LLM USAGE

LLMs, specifically ChatGPT and Gemini, supported the writing process in an iterative manner. We drafted paragraphs and asked the models for feedback on grammar and clarity. We then incorporated selected suggestions into the writing and repeated this process until we were satisfied with the writing.

The code developed for the experiments was written by the authors with the help of occasional code completions. The central components (e.g., the hybrid solver implementation and the greedy-router training pipelines) were implemented exclusively by the authors.

All substantive intellectual contributions, which include ideas, theorems, and analyses, are our own. LLMs were occasionally used to verify the correctness of proofs, but all proof strategies originated from the authors and relevant literature.

