# OpenReview forum: "A Greedy PDE Router for Blending Neural Operators and Classical Methods"
_ICLR.cc/2026/Conference — ICLR 2026 Conference Withdrawn Submission_

### Official Review · Reviewer_i478 · 2025-10-24

**Soundness:** 3
**Presentation:** 3
**Contribution:** 3
**Rating:** 6
**Confidence:** 3

**Summary:**

This paper introduces Greedy PDE Router, a hybrid PDE solver that dynamically selects among classical numerical solvers and neural operators at each iteration. The proposed method learns an approximate greedy routing rule that selects the solver achieving the largest immediate error reduction. Theoretically, the authors prove that the greedy rule provides a constant-factor approximation to the optimal strategy under weak sequence supermodularity. They train a Bayes-consistent surrogate loss to approximate this rule without access to true errors. Experiments on Poisson and Helmholtz equations show faster and more stable convergence than both single solvers and HINTS.

**Strengths:**

1. The paper moves beyond fixed-schedule hybrids like HINTS, allowing data-dependent solver selection at each iteration.

2. The paper establishes weak sequence supermodularity and provides approximation guarantees for the greedy policy.

**Weaknesses:**

1. There is limited PDE diversity. Evaluations are confined to linear elliptic PDEs on simple domains; it is unclear how well the method generalizes to nonlinear or time-dependent PDEs.

2. The router requires supervised data with reference solutions. This may not be practical for large or real-time systems.

3. The paper has limited comparison. It only compared to HINTS and single solvers; other hybrid or learned PDE solvers (e.g., operator learning + multigrid co-training) are missing.

**Questions:**

I have several questions:
1. How does the greedy router perform on nonlinear or time-dependent PDEs where the operator changes across steps?

2. Does the method remain stable when neural operator predictions are inaccurate or out-of-distribution?

3. How sensitive is the performance to the quality of training data or the initial solver ensemble?

4. Can the authors provide results comparing to more hybrid and neural operators?

---

### Official Review · Reviewer_rmmL · 2025-10-31

**Soundness:** 2
**Presentation:** 3
**Contribution:** 1
**Rating:** 2
**Confidence:** 4

**Summary:**

Authors proposed a particular learned iterative method for sparse linear systems originated from finite-difference discretisation of PDEs (stationary diffusion, Helmholtz, etc). An approach by authors is to train LSTM that selects among several available solutions methods. Methods may be classical (Jacobi, Gauss-Seidel) or learned (surrogates based on DeepONet, FNO). The training is supervised with a greedy strategy used as a target.

The use of greedy strategy is justified by theoretical results, proven by authors.

The scheme is evaluated experimentally on several linear systems in $D=1$ and $D=2$.

**Strengths:**

Iterative solution of large sparse linear systems is a mature field in numerical linear algebra with countless applications in scientific computing. Most techniques, especially relaxation techniques, are studied from the standpoint of asymptotic results. In contrast, authors of the present contribution are focused on the performance in the finite horizon, which is both novel and potentially

**Weaknesses:**

In my view, the results the authors demonstrate are not convincing. Experimental results are especially doubtful. On the theoretical side authors were unable to obtain strong results for transient behaviour.

I will clarify these points in the section below.

**Questions:**

**Problems with theoretical results**

Suppose matrix $A$ is symmetric positive definite and we have two relaxation methods with symmetric error propagation matrices that have spectral radii $0 < \rho_1 \ll \rho_2 < 1$. So both methods are convergent and one of them is much better than the other one.

An artificial requirement to have a symmetric matrix is needed to use Proposition 4.2 where the Lipschitz constant of the iterative method is used (not used normally for iterative methods).

Now, the optimal solution is to use the best method all the time, since it is much better than the second one by assumption. Given that, Proposition 4.2 implies $\alpha(O) = \max\left(\frac{1}{T}\frac{4}{1 - \rho_1^2}, 1\right)$. Roughly speaking after $5$ iterations we have $\alpha(O) = 1$ (basically for most situations, not specifically for this one). I will assume that $T\gg 1$ and we can safely say take $\alpha(O) = 1$.

Applying this to Theorem 4.1 we obtain that error (5) for greedy strategy is bounded from above
$$
\text{error}\_{\text{greedy}} \leq (1 - e^{-1})\text{error}\_{\text{optimal}} + e^{-1} \text{error}\_{\text{initial}}.
$$
Now, the error of optimal selection can be estimated in my case and it is rapidly reaching zero. The second term is still there no matter what.

I have the following questions to the authors:
1. Am I missing something?
2. If not, what is the use of such an upper bound?

**Problems with empirical evaluation**
1. The Poisson equation $-\Delta u(x) = f(x)$ with periodic boundary conditions has no unique solution even if $f(x)$ is orthogonal to the kernel of a discretised matrix. It is a very unusual choice for the benchmark of classical iterative methods and learned router. Probably because of that choice, for tiny linear systems authors use, multigrid requires over $100$ iterations to produce error $10^{-2}$. I expect that geometric multigrid results in relative error $10^{-2}$ in about $2$ or $3$ iterations, certainly not $100$. Can the authors report results for well-posed linear problems?
2. For the Helmholtz equation how $a(x)$ was generated? What is a typical value of this field? Did the authors try to use some iterative solvers tailored to the Helmholtz equation specifically?
3. Can the authors share the greedy strategy used to train their router and the strategy that the router learned? One way to do that is to use a colour coding scheme: assign colour to each solver, draw selected policy as a set of coloured squares. It would be interesting to look at those pictures for many instances for each equation. My hypothesis is that the selection of the router is rather mundane with DeepONet selected in the first few iterations and best classical method selected most of the time afterwards.

---

### Official Review · Reviewer_LCbD · 2025-11-01

**Soundness:** 3
**Presentation:** 3
**Contribution:** 3
**Rating:** 4
**Confidence:** 3

**Summary:**

This paper treats hybrid PDE solvers, which is combining classical iterative methods (Jacobi/GS/MG) with Neural Operators (e.g., DeepONet/FNO),as a sequential decision (routing) problem. At each iteration, a greedy router selects the update operator that is expected to maximally reduce the error. Under explicit assumptions for linear PDEs (e.g., contraction and zero-preserving error propagation maps), the authors establish a constant-factor approximation guarantee for the greedy selection. To address the practical challenge that the true error is unobservable, they propose training the router with pseudo-labels generated from greedy rollouts and a cost-sensitive convex surrogate loss (with Bayes consistency). On 1D/2D Poisson/Helmholtz problems, the method consistently outperforms both single-solver baselines and fixed-schedule hybrids (HINTS) on final error and error AUC, notably by skipping NO updates when counterproductive, thereby avoiding sawtooth-like regressions.

**Strengths:**

The paper articulates a principled view of hybrid PDE solving by casting the choice among classical updates and Neural-Operator updates as a sequential decision problem, and it supports this view with a constant-factor guarantee for greedy selection under natural assumptions.
The learning procedure is well aligned with the decision objective: pseudo-labels from greedy rollouts and a cost-sensitive convex surrogate ensure that the trained router optimizes for error reduction rather than a proxy that might diverge from actual risk.
The empirical results are persuasive. By skipping Neural-Operator steps when they would be harmful, the router stabilizes convergence and improves not only the final error but also the error AUC, which is operationally meaningful.
The framework is readily extensible. Because candidate updates can be swapped or augmented without redesigning the method, the approach is promising as a meta-preconditioner or meta-scheduler for a broader class of PDE solvers.

**Weaknesses:**

Experiments are restricted to 1D/2D Poisson/Helmholtz with periodic boundary conditions. There is no evaluation under non-periodic boundary conditions (Dirichlet/Neumann/mixed, including Robin). Consequently, behavior near boundaries is unclear, and it is difficult to assess whether the reported advantages in final error and error AUC generalize. Please add results on representative 1D/2D problems with these boundary conditions.

**Questions:**

1. Beyond periodic domains, do the improvements hold under Dirichlet/Neumann/mixed (Robin) boundary conditions in 1D/2D?
2. On identical hardware and iteration budgets, please provide wall-clock, approx. FLOPs, and peak memory versus baselines.

---

### Official Review · Reviewer_jj9u · 2025-11-03

**Soundness:** 3
**Presentation:** 3
**Contribution:** 3
**Rating:** 4
**Confidence:** 4

**Summary:**

This paper proposes to learn a router to adpatively chooose a solver for each iteration when running iterative solvers. Instead of inserting a neural solver in the prior work HINTS, the proposed method trains a learnable router that is approximating the solution obtained by greedy strategy. The proposed method has theoretical guarantees to approximate the greedy solutions given the true errors are known.

**Strengths:**

1. The proposed method provides a theoretical guarantee for approximating the greedy algorithm solution. The proposed surrogate loss is theoretically backed as an upper bound to the original loss.
2. The paper shows convincing results over baselines including HINTS and solver only approach. For the solvers, the paper uses different methods including Jacobi, GS and MG.
3. The proposed method may be general and could be extended to other control problems or optimization problems beyond PDEs, which shows a broader impact.

**Weaknesses:**

1. From Fig. 1, for 2D poisson, the proposed method achieves a lower error after 300 or 100 iterations than the solver only method. At iteration 1, the proposed method starts with a lower error than the solver method (due to that at the first iterations, the error reduction is larger for DeepONet than the classical solver) . But if I look at the iterations after 20 or so, the slope for the two methods are identical. This might suggest that the convergence rate might be governed by the classical solver, rather than being significantly accelerated by the routing method. Based on my observations, is it possible that for the first few iterations, the routing mainly selected the DeepONet, and then for the rest, the router mainly selected the classical solvers indicated by the parallel lines.
2. From Fig. 1, it looks like the solver only method is not bad. By giving it more iterations, it could reach the same error as the proposed method. Because of this, I can’t see a big advantage of the proposed method over the solver only method, given that the proposed method requires lots of training time for preparing a router model. But obviously, the proposed method fixes the issue of the prior work HINTS — wrong choice hurts the performance. However, it looks like the error jump comes from the DeepONet at each fixed iteration. Does it mean the best option without training a router is to first use HINTS for a few iterations, and then do an early stopping or fully use classical solvers only?

**Questions:**

1. Why at the first iteration, HINTS and solver only method don’t have the same starting errors?
2. Could the authors list the choices by the routers for each step?
3. For the HINTS and for ther iteration where the error increases, it should be the iteration for the DeepONet, right?
4. I would like the authors to answer the weakness 2.
5. Have the authors tried to incorporate more neural operators in the router choice list?

---

### Note · Authors · 2025-11-20

**Comment:**

We would like to thank the reviewers for taking the time to read our submission. We sincerely appreciate the thoughtful comments, especially the questions regarding the practical value of the router and whether the learned strategy meaningfully adapts across samples.

After careful consideration, we have decided to withdraw the paper to substantially revise the work. Your feedback has been instrumental in helping us identify key limitations and clarify the contributions. We are grateful for your constructive critiques, and they will directly shape the next iteration of this project.

Thank you again for your time, expertise, and the helpful direction you provided.

**Withdrawal Confirmation:**

I have read and agree with the venue's withdrawal policy on behalf of myself and my co-authors.